# Optimal thresholds and algorithms for a model of multi-modal learning in high dimensions

## Abstract

This work explores multi-modal inference in a high-dimensional simplified model, analytically quantifying the performance gain of multi-modal inference over that of analyzing modalities in isolation. We present the Bayes-optimal performance and weak recovery thresholds in a model where the objective is to recover the latent structures from two noisy data matrices with correlated spikes. The paper derives the approximate message passing (AMP) algorithm for this model and characterizes its performance in the high-dimensional limit via the associated state evolution. The analysis holds for a broad range of priors and noise channels, which can differ across modalities. The linearization of AMP is compared numerically to the widely used partial least squares (PLS) and canonical correlation analysis (CCA) methods, which are both observed to suffer from a sub-optimal recovery threshold.

## 1 Introduction

Multi-modal, multi-view or multi-omic data analysis and learning represent a frontier of significant complexity and potential. These approaches are characterized by their integration of diverse data types, each offering a unique perspective or 'view' on the latent phenomena under study. This integration poses two fundamental questions:

- Firstly, how can information from different modalities or views be optimally combined?

- Secondly, how much can be gained by multi-modal learning over analysis of the modalities in isolation?

Multi-modal learning in current ML focuses on learning different complex non-linear models of the modalities which ideally cross-inform each other (Ngiam et al., 2011; Baltrušaitis et al., 2018; Bayoudh et al., 2022).

In this work, we adopt a reductionist approach and study a simple linear model of multi-modal learning. This allows us to answer the two questions posed above, at least in the simple setting under consideration. In particular, our model captures the issues of (i) how much statistical power is gained by combining information from the modalities, (ii) aligning the correlated latent structures, and (iii) dealing with different priors and noise models of the modalities.

The data model we study is also underlying methods known under the name projection to latent structures (PLS) (Wold, 1975; 1983; Wegelin, 2000), originally referred to as partial least squares (PLS) in the literature, and the more broadly known canonical correlation analysis (CCA) (Hotelling, 1936) subsumed by PLS. These are linear spectral algorithms widely used in chemometrics (Wold et al., 2001; Mehmood et al., 2012), econometrics (Hulland, 1999), neuroscience (Krishnan et al., 2011) and other fields to practically solve linear multi-view inference or prediction tasks in high dimensions.

We provide a typical-case analysis of the Bayes-optimal performance in the high-dimensional limit of the model, based on approximate message passing (AMP) (Donoho et al., 2009; Zdeborová & Krzakala, 2016) with its associated low-dimensional state evolution (SE) (Bayati & Montanari, 2011; Zdeborová & Krzakala, 2016), and the associated Bethe free-energy. This analysis results in the weak recovery threshold that appears as phase transitions in the performance of AMP in the high-dimensional limit. This threshold coincides

with the weak-recovery threshold in Bayes-optimal performance if the phase transition is continuous and instead is conjectured to give the optimal performance for polynomial time algorithms in the presence of a first-order transition (Zdeborová & Krzakala, 2016). In the latter case, the Bayes-optimal threshold is determined from the Bethe free energy of the model. We also numerically demonstrate the generally good performance but sub-optimal recovery threshold of PLS even for Gaussian noise channels and priors. For completeness, we show numerical results also for CCA, which is known to have a number of disadvantages compared to "mode-A" PLS (Wegelin, 2000), and is also found here to have less favorable performance and recovery threshold.

## 1.1 Spiked Multi-modal Model

We consider the following rank-1 model with Gaussian additive noise

$$X_{ij} = \frac{\lambda_X}{\sqrt{n_X}} w_i^X v_j^X + \xi_{ij}^X \tag{1}$$

$$Y_{ij} = \frac{\lambda_Y}{\sqrt{n_Y}} w_i^Y v_j^Y + \xi_{ij}^Y \tag{2}$$

where $w^X \in \mathbb{R}^{n_X}$, $w^Y \in \mathbb{R}^{n_Y}$, $v^{X/Y} \in \mathbb{R}^d$, and $\xi_{ij}^{X/Y} \overset{\text{iid.}}{\sim} \mathcal{N}(0, \sigma_{\xi^{X/Y}}^2)$. We assume that $w_i^X$ and $w_i^Y$ are independent, while $v_j^X, v_j^Y$ are given by a correlated joint distribution, such that $X, Y \in \mathbb{R}^{n_{X/Y} \times d}$ are noisy rank-1 matrices with correlated factors $v^{X/Y}$. In the following, the view or modality index is denoted as $z \in \{X, Y\}$ and where needed, the index of the alternate view is denoted as $\bar{z}$. We consider the high-dimensional limit $\lim_{d, n_z} \to \infty$ with scaling $\frac{d}{n_z} = \alpha_z \sim \mathcal{O}(1)$.

The model can be described as a dual-view rank-1 matrix estimation with correlated latent column space, and it is a rank-1 version of the data model fitted by PLS.

While we will mostly focus on the model as given in Equations (1) and (2), in our derivations we go beyond the additive Gaussian noise, considering more general iid. noise channels

$$P_{\text{out}}^z(z_{ij}|w_i^z v_j^z) = e^{g_z(z_{ij}, w_i^z v_j^z)} \tag{3}$$

(where again $z \in \{X, Y\}$) and general entry-wise i.i.d. priors on the projection vectors $P_w^z(w_i^z)$ with variance $\sigma_{w^z}^2$ and on the joint latent vectors $P_v(v_j^X, v_j^Y)$ with cross covariance $c_v$ and variances $\sigma_{v^{X/Y}}^2$ subsumed in the covariance matrix $\Sigma$. The posterior is given by

$$P(\{w, v\}|X, Y) = \frac{1}{\mathcal{Z}(X, Y)} \quad \prod_i P_v(v_i^X, v_i^Y) \prod_{i, \{z\}} P_w^z(w_i^z) \prod_{i, j, \{z\}} P_{\text{out}}^z(z_{ij}|w_i^z v_j^z). \tag{4}$$

We aim to analyze the Bayes-optimal estimation when the priors and noise channels are assumed to match those of the ground-truth model. Note that the model has a $\mathbb{Z}_2$ symmetry, being invariant under $\{w, v\} \to \{-w, -v\}$.

Defining $S_{ij}^z = \partial_a g_z(z_{ij}, a)|_{a=0}$ and $R_{ij}^z = (\partial_a g_z(z_{ij}, a)|_{a=0})^2 + \partial_a^2 g_z(z_{ij}, a)|_{a=0}$, we assume the channel can be expanded as

$$e^{g_z(z_{ij}, w_i^z v_j^z)} = \exp\left(g_z(z_{ij}, 0) + S_{ij}^z \frac{\lambda_z}{\sqrt{n_z}} w_i^z v_j^z + \frac{1}{2}(R_{ij}^z - (S_{ij}^z)^2)\frac{\lambda_z^2}{n_z}(w_i^z v_j^z)^2 + \mathcal{O}(n_z^{-\frac{3}{2}})\right) \tag{5}$$

and we can work with general $S^z, R^z$. To recover the additive Gaussian noise case, use $S_{ij}^z = \sigma_{\xi^z}^{-2} z_{ij}$ and $R_{ij}^z = \sigma_{\xi^z}^{-4} z_{ij}^2 - \sigma_{\xi^z}^{-2}$.

We chose a rank-1 model since we believe it already captures the fundamental phenomenology of the problem. An extension to finite rank $r$ would, in analogy to single-view matrix factorization (Rangan & Fletcher, 2012; Lesieur et al., 2017), yield an additional index in the equations while the location of the phase transition for the strongest signal direction will not change. Qualitatively different behavior could appear in other scaling limits, e.g. if the signal rank is not finite but proportional to $n_z$ and $d$.

Note that the signal scales weakly as $n_z^{-1/2}$ compared to the $\mathcal{O}(1)$ noise. This is the right scaling to see the Baik-BenArous-Péché (BBP) transition of the largest singular values correlated to the rank-1 signals disappearing in the random bulk spectra of $X$ and $Y$ at (for unit variances) $\lambda_z = \alpha_z^{-\frac{1}{4}}$ (Benaych-Georges & Nadakuditi, 2012). We will quantify the improvement that comes from exploiting the correlation between $v^X$ and $v^Y$ over the BBP thresholds of the two modalities in isolation.

## 1.2 Related work

A large number of practical methods for linear multi-view data analysis have been proposed which we do not review in detail. We compare against PLS (Wold, 1975) which exists in several variants (Wegelin, 2000; Rosipal & Krämer, 2006). Notably CCA is equivalent to "mode-B" PLS, but despite its broad popularity is well known for severe shortcomings compared to the canonical "mode-A" PLS (Wegelin, 2000) which we therefore consider instead. These methods are based on the singular value spectrum of the correlation matrix $XY^T$ in the case of PLS ("mode-A") and that of the normalized correlation matrix $(XX^T)^{-\frac{1}{2}}XY^T(YY^T)^{-\frac{1}{2}}$ in the case of CCA.

The canonical PLS algorithm finds rank-k approximations of $X$ and $Y$ by iterating k times the steps: 1) computing the top pair $s_X, s_Y$ of singular vectors of $XY^T$, 2) estimating $\hat{v}_z = Z^T s_z$, 3) finding refined estimates $\hat{w}_z$ by regressing $Z$ on $\hat{v}_z$ so that $\hat{w}_z = (\hat{v}_z^T \hat{v}_z)^{-1} Z \hat{v}_z$, 4) subtracting the rank-1 approximations obtained from each data matrix in isolation, $Z \leftarrow Z - \hat{w}_z \hat{v}_z^T$, 5) repeat from 1). As a simplified variant, PLS-SVD eschews steps 3) and 4), only computing the singular vectors of $XY^T$ as the estimates $\hat{w}'_z = s_z$ and again $\hat{v}'_z = Z^T s_z$. After the first iteration, which is the only one required in our rank-1 setting, the two variants only differ in that $\hat{w}'_z = s_z$ for PLS-SVD while $\hat{w}_z = (\hat{v}_z^T \hat{v}_z)^{-1} ZZ^T s_z$ for PLS-Canonical. The weak recovery thresholds of both variants are thus the same since these estimates only have nonzero overlap with the ground-truth signals $w_z$ if the spectrum of $XY^T$ has an outlier singular value correlated with the signal.

While the spectrum of $XY^T$ has, to our knowledge, not been studied analytically, recent mathematical works exist for the spectrum and BBP-type transition of the normalized correlation matrix in CCA (Bao et al., 2019; Yang, 2022; Bykhovskaya & Gorin, 2023). We show in Figure 3 that the threshold and performance of CCA can be quite far from those of PLS and from the Bayes-optimal values. Empirically, the benefit of shared dimensionality reduction through PLS or CCA compared to single-view methods was analyzed by Abdelaleem et al. (2023), although in a different scaling regime with a stronger signal compared to ours. Non-linear and deep generalizations of CCA have also been developed in the context of self-supervised learning (Balestriero et al., 2023).

The framework we employ is based on a recently matured literature on the statistical physics of algorithmic hardness and Bayes optimal inference (Mézard & Montanari, 2009; Zdeborová & Krzakala, 2016), many aspects of which have now been made rigorous (Bayati & Montanari, 2011; Bolthausen, 2014; Celentano et al., 2020; Krzakala et al., 2023). In particular, we follow largely the notation of Lesieur et al. (2017), who analysed in detail and along related lines a single-view version of the model considered here.

While the single-view spiked matrix model has been studied intensely, e.g. (Rangan & Fletcher, 2012; Lesieur et al., 2017; Montanari & Venkataramanan, 2021), works analysing recovery thresholds for systems that can be related to multi-view or multi-modal learning have so far mainly focused on regression with side information and on variants of community detection. First we note that in mixed matrix-tensor models with rank-1 spike (Sarao Mannelli et al., 2020) the matrix information can be seen as a second view of the rank-1 signal which aids its detection in the tensor data. Kadmon & Ganguli (2018) have applied the AMP framework to low-rank tensor decomposition, where the higher-order tensor can be thought of as data matrices from an experiment with multiple varying conditions forming the additional axes. Compared to our model, this corresponds to more than two views, the rank-1 signals of which only differ by a scalar factor for each additional axis, and no difference in priors is allowed. Rigorous results on AMP for linear regression with side information have been presented by Liu et al. (2019) where the side information is a noisy version of the signal, and by Nandy & Sen (2023) where the side information is given by correlations of signal entries. Chen et al. (2018; 2022) analysed a data matching setting where both views have the same number of features and differ only by their noise realization and a permutation of the feature indices. Deshpande et al. (2018) presented the contextual stochastic block model. Recently an extension was analyzed by Duranthon & Zdeborová (2023), and we note

a line of ongoing rigorous work on AMP in multi-view variants of community detection in stochastic block models (Ma & Nandy, 2023; Yang et al., 2024). In these works one view is always a square matrix given by the adjacency matrix of a graph. This is in contrast to our model which can be interpreted as observing an arbitrary number of samples from two views of a correlated latent structure, so that both data matrices are rectangular with $n_{X/Y}$ features and $d$ samples, which allows us to compare with PLS and CCA.

### 1.3 Main contributions

- The information-theoretic performance limits for the multi-view inference task (4), obtained from the state evolution of AMP and the Bethe free-energy.

- A quantification of the signal-to-noise gain from optimally combining two views, given the prior assumption of a covariance $c_v$ between the latent vectors. E.g. for $c_v = 0.8$ and otherwise unit parameters, recovery is possible from $\sigma_{\xi^z} \approx 1.13$, compared to $\sigma_{\xi^z} = 1$ for the single-view case. We also demonstrate the distance of the recovery threshold of CCA known from Bykhovskaya & Gorin (2023) to the Bayes-optimal value.

- A spectral method with optimal sensitivity as a linearization of AMP, which combines information from the individual and correlated view, and its comparison to PLS and CCA that both result in sub-optimal sensitivity.

## 2 Approximate message passing and state evolution

### 2.1 AMP

In this section, we discuss the conceptual steps leading from belief propagation (BP) to the AMP algorithm. The main technical contributions here are the formulation of a parsimonious multi-view model in Section 1.1 and the treatment of correlated latent variables by two-dimensional marginals in the BP messages. The remaining derivation of AMP and the state evolution then goes through as a straight-forward generalization of the calculations for single-view matrix factorization presented by Lesieur et al. (2017), whose notation we adapt slightly for more consistency with standard symbols in statistical physics. The technical derivation is given in Appendices A and B.

The factor graph of the model is given in Figure 1, corresponding to the BP equations

$$m_{i \to ij}^z(w_i^z) = \frac{P_w^z(w_i^z)}{\mathcal{Z}_{i \to ij}^{z,m}} \prod_{k \neq j}^d \tilde{m}_{ik \to i}^z(w_i^z) \tag{6}$$

$$\tilde{m}_{ij \to i}^z(w_i^z) = \int \frac{\mathrm{d}v_j^X \mathrm{d}v_j^Y}{\mathcal{Z}_{ij \to i}^{z,m}} \, n_{j \to ij}^z(v_j^X, v_j^Y) P_{\mathrm{out},ij}^z \tag{7}$$

$$n_{j \to ij}^z(v_j^X, v_j^Y) = \frac{P_v(v_j^X, v_j^Y)}{\mathcal{Z}_{j \to ij}^{z,n}} \prod_{k \neq i}^{n_z} \tilde{n}_{kj \to j}^z(v_j^z) \times \prod_k^{n_{\bar{z}}} \tilde{n}_{kj \to j}^{\bar{z}}(v_j^{\bar{z}}) \tag{8}$$

$$\tilde{n}_{ij \to j}^z(v_j^z) = \int \frac{\mathrm{d}w_i^z}{\mathcal{Z}_{ij \to j}^{z,n}} \, m_{i \to ij}^z(w_i^z) P_{\mathrm{out},ij}^z. \tag{9}$$

Again $\bar{z}$ refers to the opposite modality compared to $z$. Note that we treat $v_j^X, v_j^Y$ as a joint variable such that $n_{j \to ij}^z(v_j^X, v_j^Y)$ is a two-dimensional marginal. As a consequence, the message distribution is additionally being marginalized over the unused variable in Equation (7), as e.g. $P_{\mathrm{out},ij}^X$ depends only on $v^X$. This leads to a more parsimonious notation than introducing additional messages with a factor representing the correlation of both variables, and is nothing else than what is conventionally done with the index dimension of vectors with iid. priors such as for $m_{i \to ij}^z(w_i^z)$. The vector $w^z$ can also be seen as a joint variable and the associated message factorizes with the marginalization over all $w_{k \neq i}^z$ implicit, due to the iid. prior. In the presence of a correlated prior the underlying perspective of joint variables becomes relevant since the joint prior appears in Equation (8); while if $P_v(v_j^X, v_j^Y)$ would factorize, also the message $n_{j \to ij}^z(v_j^X, v_j^Y)$ would factorize.

In the high-dimensional limit $d \to \infty$, while the messages do not become Gaussian for arbitrary priors, exploiting the noise channel expansion (5) the BP iteration closes on the means and variances of the messages.

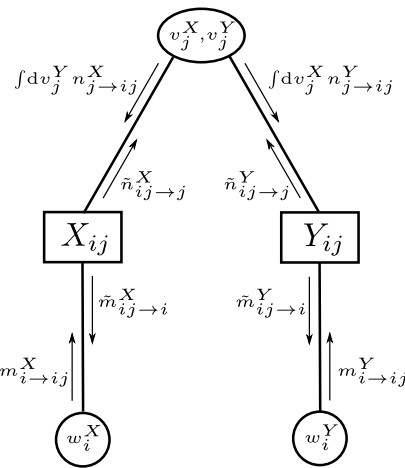

Figure 1: Factor graph of the model. Note that the index dimension is implicit while the $X/Y$ dimension has been emphasized because the latent variables $v_j^X, v_j^Y$ have a correlated prior. Yet the principle remains the same: In a message on an edge $\{X/Y, ij\}$ all other dimensions are marginalized.

The resulting iteration on means and variances instead of distributions is called relaxed belief propagation (rBP).

The form of the underlying marginal distributions becomes that of a tilted prior distribution

$$\mathcal{W}(x, K, J) = P_x(x) \exp(Jx - \frac{1}{2}x^T K x) \tag{10}$$

where $x \in \mathbb{R}$ for $m_{i \to ij}^z(w_i^z)$ and $x \in \mathbb{R}^2$ for $n_{j \to ij}^z(v_J^X, v_j^Y)$. We also define the normalization of this distribution as $\mathcal{Z}(K, J) = \int dx\, \mathcal{W}(x, K, J)$, which appears again in the free energy, Appendix D.3. Interpreting $J$ as a linear source term of the cumulant-generating function $\log \mathcal{Z}(K, J)$ of $x \sim \mathcal{W}(K, J)$, we can write the mean and variance as derivatives w.r.t. the source terms. In compliance with standard notation, we introduce the first derivative (the mean) as the "denoising" function

$$f_{\text{in}}^x(K, J) = \frac{\partial}{\partial J} \log \int dx\, \mathcal{W}(x, K, J). \tag{11}$$

In the case of $v^z$ the off-diagonal terms of $K$ never appear, thus we simplify the notation to $f_{\text{in}}^{v^z}(K_X, K_Y, J_X, J_Y)$ where the $z$ index results from taking the derivative by $J_X$ or $J_Y$, respectively. However, the term "denoising function" should not obscure the fact that $f_{\text{in}}^x(K, J)$ and $\frac{\partial f_{\text{in}}^x}{\partial J}(K, J)$ are by definition nothing but the first and second cumulants of the marginal density at the next time step, given by the tilted prior $\mathcal{W}(x, K, J)$.

From rBP (A.14)-(A.17) which is based on the $\mathcal{O}(d^2)$ messages on the edges of the factor graph, we then obtain AMP which iterates $\mathcal{O}(d)$ node-specific estimates by exploiting that the dependence of the rBP estimates on the target index is weak and can be discounted for by the Onsager reaction term with appropriately delayed time index (Bolthausen, 2014). Concerning the update order determining the time indices of the AMP iteration, while conventionally all messages are passed and updated synchronously for simplicity, there is a freedom to choose an arbitrary update order. Here we choose to update the messages in two sequential blocks, first the marginals of $v^z$, then those of $w^z$. This is to avoid limit cycles of length 2 arising from the $\mathbb{Z}_2$ symmetry in the problem if the relative sign of the $w$ and $v$ estimates does not match. For example, for vanishing noise the otherwise perfect estimate $-w^z, v^z$ would be updated to $w^z, -v^z$ and back to $-w^z, v^z$,

---

**Algorithm 1** AMP (v-first)

---

**Input:**

data $X, Y$

parameters $\alpha_z, \lambda_z, \sigma_{\xi^z}, \Sigma_v, \sigma_{w^z}$ for $z \in \{X, Y\}$

**Initialize:**

$\hat{\sigma}_w^z, \hat{\sigma}_{v^z}^z \leftarrow \sigma_{w^z}^2, \sigma_{v^z}^2$

$\hat{v}^z \leftarrow 0$

**if** "approx. Nishimori" **then**

    $\hat{w}^z \leftarrow \frac{w_p^z}{\sqrt{n_z}}$ with sample $w_p^z \sim P_{w^z}$

**else if** "informed" **then**

    $\hat{w}^z \leftarrow w_0^z$

**else if** "spectral" **then**

    $\hat{w}^z \leftarrow \text{Poweriter}(\Gamma_w)$

**end if**

**Run:**

**while not converged do**

    # update v sector first

    $K_z^v, J_z^v \leftarrow$ Equations (12) and (13)

    $\hat{v}^z \leftarrow f_{\text{in}}^{v^z}(K_X^v, K_Y^v, J_X^v, J_Y^v)$

    $\hat{\sigma}_v^z \leftarrow \frac{\partial f_{\text{in}}^{v^z}}{\partial J_z^v}(K_X^v, K_Y^v, J_X^v, J_Y^v)$

    # update w sector second

    $K_z^w, J_z^w \leftarrow$ Equations (14) and (15)

    $\hat{w}^z \leftarrow f_{\text{in}}^w(K_z^w, J_z^w)$

    $\hat{\sigma}_w^z \leftarrow \frac{\partial f_{\text{in}}^w}{\partial J_z^w}(K_z^w, J_z^w)$

**end while**

**return** $\hat{w}^z, \hat{\sigma}_{w^z}, \hat{v}^z, \hat{\sigma}_{v^z}$ for $z \in \{X, Y\}$

---

etc. The source terms determining the AMP iteration are then

$$J_{z,j}^{v,t} = \frac{\lambda_z}{\sqrt{n_z}} \sum_k^{n_z} S_{kj}^z \hat{w}_k^{z,t-1} - \frac{\lambda_z^2}{n_z} \hat{v}_j^{z,t-1} \sum_k^{n_z} (S_{kj}^z)^2 \hat{\sigma}_{w,k}^{z,t-1} \tag{12}$$

$$K_{z,j}^{v,t} = \frac{\lambda_z^2}{n_z} \sum_k^{n_z} \left[ (S_{kj}^z \hat{w}_k^{z,t-1})^2 - R_{kj}^z ((\hat{w}_k^{z,t-1})^2 + \hat{\sigma}_{w,k}^{z,t-1}) \right] \tag{13}$$

$$J_{z,i}^{w,t} = \frac{\lambda_z}{\sqrt{n_z}} \sum_k^d S_{ik}^z \hat{v}_k^{z,t} - \frac{\lambda_z^2}{n_z} \hat{w}_i^{z,t-1} \sum_k^d (S_{ik}^z)^2 \hat{\sigma}_{v,k}^{z,t} \tag{14}$$

$$K_{z,i}^{w,t} = \frac{\lambda_z^2}{n_z} \sum_k^d \left[ (S_{ik}^z \hat{v}_k^{z,t})^2 - R_{ik}^z ((\hat{v}_k^{z,t})^2 + \hat{\sigma}_{v,k}^{z,t}) \right] \tag{15}$$

We would like to point out that the time indices $t-1$ for both Onsager reaction terms in (14) and (12) are correct because for the sequential update order, $\hat{v}_j^{z,t}$ is updated based on $\hat{w}_i^{z,t-1}$ while $\hat{w}_i^{z,t}$ is updated based on $\hat{v}_i^{z,t}$.

## 2.2 Linearized AMP

The AMP algorithm assumes knowledge of the parameters of the model and the corresponding priors. While these can be learned in practice via expectation maximization procedures it is also beneficial to derive spectral algorithms that require fewer assumptions. A standard way toward these is linearization of AMP around its trivial fixed point as done e.g. in Krzakala et al. (2013).

In Appendix C, instead of directly expanding AMP (Algorithm 1) for small mean estimates $\hat{w}^z, \hat{v}^z \ll 1$ which would give an undesirable non-Markovian dependence on past iterates through the Onsager reaction term, we expand the rBP equations (A.14)-(A.17) and then, calculating the appropriate Onsager correction, do the step from linearized rBP to the linearized AMP power-iterations

$$\hat{v}^t = \Gamma_v \hat{v}^{t-1} \qquad\qquad\qquad \hat{w}^t = \Gamma_w \hat{w}^{t-1} \tag{16}$$

where the notation without modality index $z$ signifies the stacked vector, $\hat{v}^t = \left(\hat{v}_1^{X,t}, ..., \hat{v}_d^{X,t}, \hat{v}_1^{Y,t}, ..., \hat{v}_d^{Y,t}\right)^T \in \mathbb{R}^{2d}$ and $\hat{w}^t \in \mathbb{R}^{n_X + n_Y}$. Also we have split the iteration alternating between $v$ and $w$ sectors into two self-contained iterations with block-structured linear operators

$$\Gamma_v = \begin{pmatrix} \frac{\lambda_X^2}{n_X} \sigma_{v^X}^2 \sigma_{w^X}^2 S_X^T S_X & \frac{\lambda_Y^2}{n_Y} c_v \sigma_{w^Y}^2 S_Y^T S_Y \\ \frac{\lambda_X^2}{n_X} c_v \sigma_{w^X}^2 S_X^T S_X & \frac{\lambda_Y^2}{n_Y} \sigma_{v^Y}^2 \sigma_{w^Y}^2 S_Y^T S_Y \end{pmatrix} - \text{diag} \tag{17}$$

and

$$\Gamma_w = \begin{pmatrix} \frac{\lambda_X^2}{n_X} \sigma_{v^X}^2 \sigma_{w^X}^2 S_X S_X^T & \frac{\lambda_x \lambda_Y}{\sqrt{n_X n_Y}} c_v \sigma_{w^X}^2 S_X S_Y^T \\ \frac{\lambda_x \lambda_Y}{\sqrt{n_X n_Y}} c_v \sigma_{w^Y}^2 S_Y S_X^T & \frac{\lambda_Y^2}{n_Y} \sigma_{v^Y}^2 \sigma_{w^Y}^2 S_Y S_Y^T \end{pmatrix} - \text{diag} \tag{18}$$

where the linear Onsager correction $-\text{diag}$ amounts to setting the diagonal to zero. The form is true for general zero-mean priors and noise channels. For completeness, the pseudo-code for the linearized AMP iteration is given in Appendix C.

Since $S_z^T S_z \hat{v}^z$ gives an estimate of the top right singular vector of $S_z$, $S_z S_z^T \hat{w}^z$ that of the top left singular vector, and $S_z^T S_{\bar{z}} \hat{w}^{\bar{z}}$ again an estimate of the top left singular vector of $S_z$ if the top right singular vectors of $S_X$ and $S_Y$ are correlated, we see that running the power-iterations Equations (17) and (18) amounts to estimating the top pair of singular vectors of the Fisher score matrices $S_X, S_Y$, which are proportional to the data matrices $X, Y$ in the Gaussian noise case.

How can we relate this linearized AMP algorithm to canonical spectral methods such as PLS? PLS works on the correlation matrix $XY^T$ while linearized AMP combines an estimate from the modality itself with an estimate from the other modality. As a consequence, it is clear that PLS will have a sub-optimal recovery threshold for low correlations, since it sees the modalities only through the correlation matrix. Linearized AMP, on the other hand, combines individual and shared information, however it does so optimally for weak recovery while the performance of estimating $v_0^z$ in the presence of small noise will be sub-optimal, because as seen from (17) the very accurate estimate of $v_0^z$ based on the individual modality will be corrupted by a correlated but different estimate of $v_0^{\bar{z}}$ based on the other modality.

The nonlinear AMP iteration solves this dilemma by reweighting the blocks in the linearization $\Gamma_v(\hat{v}^X, \hat{v}^Y)$ as the norm of the estimates grows, yielding both optimal sensitivity and performance. As a consequence, even for very small noise, AMP will never converge in a single step, but require at least two steps due to the switch from weak recovery to precise estimation of the latent signal directions.

## 2.3 Limit of perfect correlation, $c_v \to 1$

If the latent vectors are perfectly correlated, $v_j^X = v_j^Y$, the structure of the model simplifies, since the rank-1 matrices can be stacked along the feature dimension to a single rank-1 matrix. At the example of additive noise, with $w = (w_X^T, w_Y^T)^T \in \mathbb{R}^{n_X + n_Y}$ and $\xi = (\xi_X^T, \xi_Y^T)^T \in \mathbb{R}^{(n_X + n_Y) \times d}$ one obtains a single data matrix $Z = wv^T + \xi$. It then follows that, while the priors and noise channels can differ across entries, the problem has been reduced to the single-view case with the two measurements of each sample stacked into one vector. In terms of the factor graph, Figure 1, the right and left branches can be folded on top of each other in the index dimension, removing the $X/Y$ dimension.

## 2.4 State evolution

By introducing a set of order parameters we now derive the low-dimensional effective dynamics of rBP in the high-dimensional limit, known as state evolution (SE). Since for $d \to \infty$ AMP tracks the dynamics of

rBP, the SE is an effective dynamics of AMP as well. Here we sketch the conceptual steps, commenting on a subtlety in applying the Nishimori identity, and give the simplified form arising for Bayes-optimal priors and Gaussian noise channel. The full derivation is detailed in Appendix D.

The starting point are the rBP equations, since in contrast to AMP, the messages of rBP are still independent. Denoting the ground-truth vectors as $w_z^0, v_z^0$ and introducing the order parameters

$$M_w^{z,t} = \frac{1}{n_z} \sum_{i \neq j}^{n_z} \hat{w}_{i \to ij}^{z,t} w_{z,i}^0 \qquad\qquad M_v^{z,t} = \frac{1}{d} \sum_{j \neq i}^{d} \hat{v}_{j \to ij}^{z,t} v_{z,j}^0 \qquad (19)$$

$$Q_w^{z,t} = \frac{1}{n_z} \sum_{i \neq j}^{n_z} \hat{w}_{i \to ij}^{z,t} \hat{w}_{i \to ij}^{z,t} \qquad\qquad Q_v^{z,t} = \frac{1}{d} \sum_{j \neq i}^{d} \hat{v}_{j \to ij}^{z,t} \hat{v}_{i \to ij}^{z,t}, \qquad (20)$$

conventionally referred to as overlaps (or magnetizations) and self-overlaps we can use that due to independence of the messages, node-averaged quantities concentrate to their mean, which is also the mean over the noise disorder. For such self-averaging quantities one can therefore replace the node average by a disorder average. Note that in (19)-(20) we already dropped the target index of the order parameters for this reason. In this way one finds that the quadratic source terms $K$ concentrate to their mean, while the linear source terms $J$ become Gaussian variables. Finally, Bayes-optimality of the priors enables the use of the Nishimori identities (Nishimori, 2001), which yield the simplification $Q_{w/v}^z = |M_{w/v}^z|$.

Here we wish to make a technical comment why the absolute value appears as a consequence of the $\mathbb{Z}_2$ symmetry being spontaneously broken by the random initialization. We believe this clarifies how to deal with this symmetry with respect to the existing literature on state evolution for similar systems, e.g. (Lesieur et al., 2017; Kadmon & Ganguli, 2018). For the Nishimori conditions to hold at all times, initialization of the mean estimators $\hat{w}^z, \hat{v}^z$ must be at zero, consistent with the mean of the prior distribution. Yet zero is a fixed point of the iteration due to symmetry. In practice, AMP is thus initialized with a small random direction, randomly breaking the symmetry and choosing the global signs between $\hat{w}^z$ and $\hat{v}^z$. Now, in words, the Nishimori identity (Nishimori, 1980; 2001) states that in a quantity averaged both over the posterior distribution, e.g. $P(w|X)$, and the disorder distribution, we can replace one of any iid. sampled variables from the posterior by a variable sampled from the prior distribution, that is

$$\mathbb{E}_{w^0} \mathbb{E}_{w_1, w_2 \sim P(w|X_{w^0})} [f(w_1, w_2, ...)]$$
$$= \mathbb{E}_{w^0} \mathbb{E}_{w_1, w_2 \sim P(w|X_{w^0})} [f(w^0, w_2, ...)] . \qquad (21)$$

However, depending on which direction the $\mathbb{Z}_2$ symmetry is broken, $\hat{w}^z$ and $\hat{v}^z$ are in fact estimators of $\pm w^z$ and $\pm v^z$. Therefore we need to replace the variable from the posterior, e.g. $\hat{w}^X$, by $\pm w^X$ depending on the sign of the overlap $M_w^X$. This results in the relation $Q_{w/v}^z = |M_{w/v}^z|$, restores the symmetry of the SE equations with respect to the sign of the overlaps, see Figure S1, and avoids the obviously erroneous situation of negative $Q_{w/v}^z$ that can arise otherwise.

With $Q_{w/v}^z = |M_{w/v}^z|$, the Bayes-optimal state evolution for Gaussian noise channel then amounts to

$$M_v^{z,t} = \mathbb{E}_{v_{X,Y}^0, J_{X,Y}^{v,t}} \left[ f_{\text{in}}^{v^z} \left( |\tilde{M}_w^{X,t-1}|, |\tilde{M}_w^{Y,t-1}|, J_X^{v,t}, J_Y^{v,t} \right) v_z^0 \right] \qquad (22)$$

$$M_w^{z,t} = \mathbb{E}_{w_z^0, J_z^{w,t}} \left[ f_{\text{in}}^w \left( \alpha_z |\tilde{M}_v^{z,t}|, J_z^{w,t} \right) w_z^0 \right] \qquad (23)$$

with $\tilde{M}_{w/v}^{z,t} = \frac{\lambda_z^2}{\sigma_{\xi z}^2} M_{w/v}^{z,t}$ and

$$J_z^{v,t} \sim \mathcal{N} \left( \tilde{M}_w^{z,t-1} v_z^0, \ |\tilde{M}_w^{z,t-1}| \right) . \qquad (24)$$

$$J_z^{w,t} \sim \mathcal{N} \left( \alpha_z \tilde{M}_v^{z,t} w_z^0, \ \alpha_z |\tilde{M}_v^{z,t}| \right) \qquad (25)$$

Refer to (D.24)-(D.33) for the form of the SE equations without Bayes-optimal priors and for general noise channels. Depending on the prior, all or part of the expectations in Equations (22) and (23) can be computed analytically, see Appendix D.2 for Gaussian and Rademacher-Bernoulli priors.

### 2.5 Algorithmic and information-theoretic weak recovery thresholds

Linearizing the SE, by plugging (22) into (23) then expanding for $M_w^z = \epsilon^z \ll 1$, we can assess the stability of the uninformative state at zero overlaps by computing the maximum eigenvalue $\eta_+$ of the resulting $2 \times 2$ matrix. For zero-mean priors, where consequently the prior overlaps are zero, the algorithmic weak recovery threshold $\theta_{\mathrm{alg}}$ is defined as the smallest signal-to-noise ratio (snr) above which AMP recovers the latent variables better than drawing from the prior. This threshold takes place when $\eta_+ = 1$. Defining the normalized correlation coefficient $\hat{c}_v = \frac{c_v}{\sigma_{vX}\sigma_{vY}}$ and the effective snr

$$\tilde{\lambda}_z = \alpha_z \lambda_z^4 \frac{\sigma_{v^z}^4 \sigma_{w^z}^4}{\hat{\Delta}_z^2}, \tag{26}$$

where $\hat{\Delta}_z$ as defined in (D.11) reduces to $\hat{\Delta}_z = \sigma_{\xi^z}^2$ for the Gaussian channel. The form of (26) arises intuitively when noting that rescaling the model (1,2) by setting the variances $\sigma_{v^z}, \sigma_{w^z}, \sigma_{\xi^z} \to 1$ corresponds to rescaling $\lambda_z \to \lambda_z \frac{\sigma_{v^z}\sigma_{w^z}}{\sigma_{\xi^z}}$, and that $\lambda_z \sim \alpha_z^{-\frac{1}{4}}$ is the scaling of the BBP transition for each single-view matrix (Benaych-Georges & Nadakuditi, 2012). We find that with zero-mean priors the algorithmic weak recovery threshold $\theta_{\mathrm{alg}}$ is given by the condition

$$1 \overset{!}{=} \frac{1}{2}\left(\tilde{\lambda}_X + \tilde{\lambda}_Y + \sqrt{\tilde{\lambda}_X^2 - 2(1 - 2\hat{c}_v^4)\tilde{\lambda}_X\tilde{\lambda}_Y + \tilde{\lambda}_Y^2}\right) \tag{27}$$

for general priors and noise channels, assuming they are Bayes-optimal. We use the $\overset{!}{=}$ notation to signify an equality condition which must be verified, here at the threshold. For symmetric $\tilde{\lambda}_X = \tilde{\lambda}_Y = \tilde{\lambda}$ (27) reduces to

$$\tilde{\lambda} \overset{!}{=} \frac{1}{1 + \hat{c}_v^2}. \tag{28}$$

In the case of perfect correlation $\hat{c}_v^2 = 1$, the simplification is

$$1 \overset{!}{=} \tilde{\lambda}_X + \tilde{\lambda}_Y \tag{29}$$

and for vanishing correlation $\hat{c}_v = 0$ we recover the threshold condition of the single-view model, $1 \overset{!}{=} \tilde{\lambda}_z$.

It is generally conjectured that no polynomial time algorithm can perform better than the $\theta_{\mathrm{alg}}$ of AMP, see Zdeborová & Krzakala (2016). For some ranges of parameters the so-called first-order phase transitions may appear in the problem as shown in Figure 2 for sparse prior on $w^z$. In those cases, the algorithmic threshold for weak recovery may not coincide with the information-theoretic threshold for weak recovery $\theta_{\mathrm{IT}}$. We define $\theta_{\mathrm{IT}}$ as: the smallest snr at which the overlap of the posterior maximum departs from the overlap achieved by the prior. This can be assessed by the Bethe free energy associated to the state evolution given in Equation (D.49). Being the negative log of the posterior, the free energy has two minima inside the spinodal regime of a first-order transition. If the lower-overlap branch is uninformative with zero overlaps, $\theta_{\mathrm{IT}}$ is given by the smallest snr where the minimum of the upper-branch becomes deeper than that of the uninformative branch.

## 3 Numerical results and phase diagram

We numerically investigate two setups with Gaussian noise channel, corresponding to Equations (1) and (2). One with both Gaussian priors on $w_i^z \sim \mathcal{N}(0, \sigma_{w^z}^2)$ and $(v_j^X, v_j^Y) \sim \mathcal{N}(0, \Sigma_v)$ with variances $\Sigma_{v,zz} = \sigma_{v^z}^2$ and covariance $\Sigma_{v,z\bar{z}} = c_v$, and in the second with the same joint Gaussian prior on the latent vectors $v^z$ but a sparse Rademacher-Bernoulli prior on $w^z$

$$P_{w^z}^{RB}(w^z) = \frac{\rho_{w^z}}{2}[\delta(w^z - 1) + \delta(w^z + 1)] + (1 - \rho_{w^z})\delta(w^z). \tag{30}$$

The corresponding denoising functions are given in Appendices B.1 and B.2.

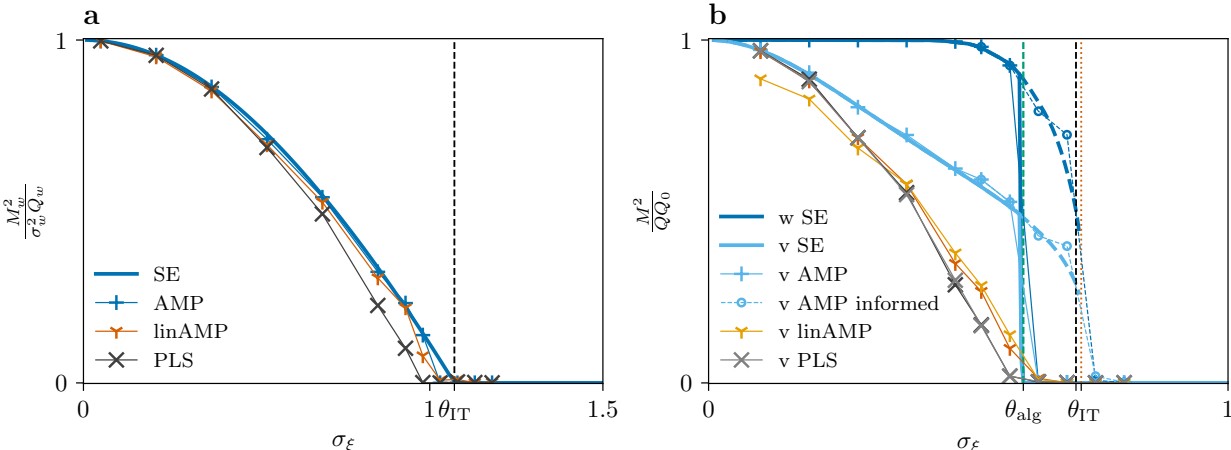

Figure 2: Phase transition of Bayes-optimal recovery (state evolution, blue lines) as a function of the noise strength, compared to AMP, PLS, linearized AMP and informed AMP. Performance is measured as the squared cosine similarity between estimated and ground-truth vectors, e.g. $\text{CS}_w^2 = \frac{M_w^2}{Q_w Q_{w^0}}$, where the square removes the arbitrary sign of the overlap arising from the $\mathbb{Z}_2$ symmetry. **a** Continuous transition for Gaussian priors on $w^z$ and $v^z$. Since the two are very close, results are shown here for $w^z$, and those for $v^z$ in Figure S3. The weak recovery threshold is $\theta_{\text{IT}} = \theta_{\text{alg}} \approx 1.07$. **b** First-order transition for Rademacher-Bernoulli (sparse) prior on $w^z$ with sparsity $\rho_z = 0.02$ and Gaussian prior on $v^z$. Lighter colors refer to $v^z$ and darker colors to $w^z$. The vertical lines are the algorithmic weak recovery threshold $\theta_{\text{alg}} \approx 0.61$ (green dashed), the information-theoretic threshold $\theta_{\text{IT}} \approx 0.71$ (black dashed) where the upper branch starts dominating the posterior based on the free energy (D.51) , and the spinodal point $\theta_{\text{sp}} \approx 0.72$ (orange dotted). Parameters are for both $z \in \{X, Y\}$: $\alpha_z = 1, \sigma_{v^z} = 1, \sqrt{c_v} = 0.75$, then for panel **a** $\lambda_z = 1, \sigma_{w^z} = 1$, and for panel **b** $\lambda_z = 4, \rho_{w^z} = 0.02$. Each algorithm performance marker is based on one run at size $d = 15000$.

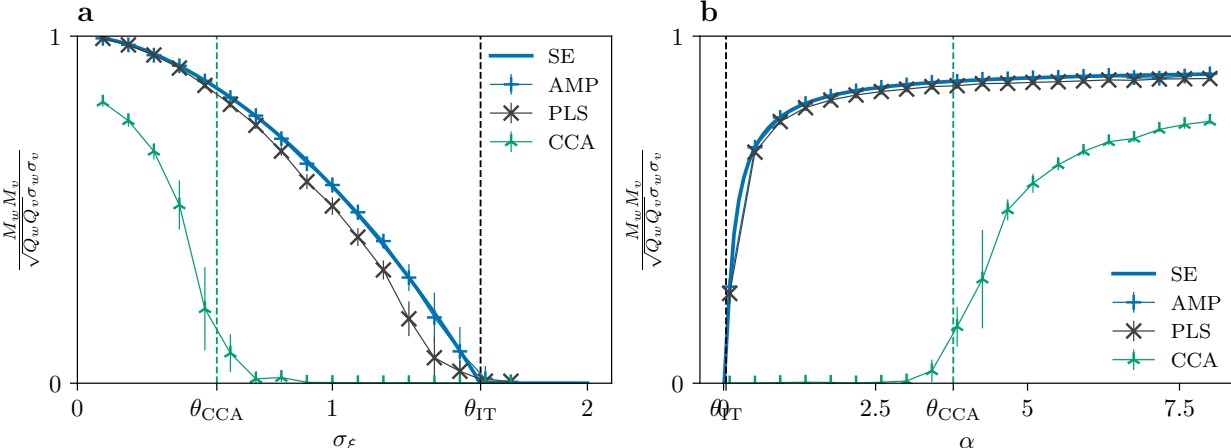

Figure 3: Phase transition and comparison of CCA (green 'λ') to PLS (gray 'x') and the Bayes-optimal performance limit for $c_v = 0.75$. Here the product of cosine similarities of the $w^z$ and $v^z$ estimates is shown for visual convenience, and the standard deviations across 10 realizations. The threshold of CCA is known analytically from theorem 2.5 of Bykhovskaya & Gorin (2023), and shown here as $\theta_{CCA}$ (green dashed), using the map of notation $\alpha^X \to \tau_K$, $\alpha^Y \to \tau_M$ and $\frac{c_v^2}{(1+\sigma_{\xi X}^2/\lambda_X^2)(1+\sigma_{\xi Y}^2/\lambda_Y^2)} \to r^2$. **a** Plot as in Figure 2a but with $\alpha^z = 4$ (other parameters $\lambda^z = \sigma_w^z = \sigma_v^z = 1$ and $d = 5000$), since CCA requires $\alpha > 1$ so that the covariance matrices $XX^T$ and $YY^T$ are invertible. The threshold of CCA ($\theta_{CCA} \approx 0.55$) is considerably lower than that of PLS. **b** Varying $\alpha^z$ instead of $\sigma_\xi^z$, here for $\lambda^z = 2$ while $\sigma_\xi^z = \sigma_w^z = \sigma_v^z = 1$ and $dn_z = 5000^2$. The threshold of CCA is $\theta_{CCA} \approx 3.78$ compared to $\theta_{\text{IT}} \approx 0.04$.

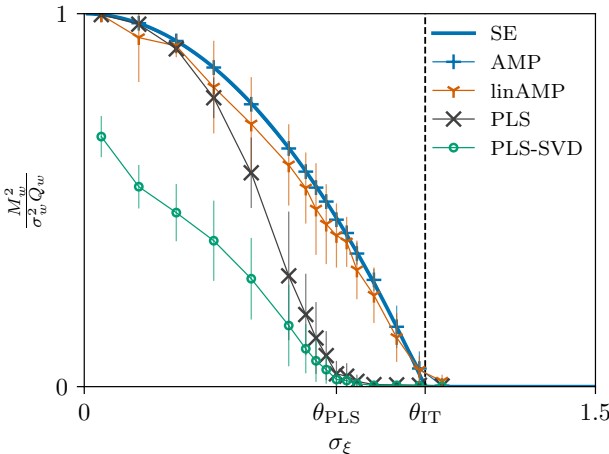

Figure 4: Phase transition and comparison to sub-optimal threshold of PLS (gray 'x') for the Gaussian prior model with smaller correlation of the latent vectors, $\sqrt{c_v} = 0.2$. Other model parameters as in Fig.2. Again results for $v^z$ are shown in Figure S4. Mean and standard deviation across 20 realizations with $d = 5000$ are shown. The threshold of PLS is estimated at $\theta_{\text{PLS}} \approx 0.74 \pm 0.03$ while $\theta_{\text{alg}} = \theta_{\text{IT}} \approx 1.01$.

In Figure 2 we compare the Bayes-optimal performance in the high-dimensional limit obtained from state evolution to the empirical performances of AMP (Algorithm 1), linearized AMP (17,18) and `PLSCanonical` from the scikit-learn library.

Note that there exist a number of variations of PLS, see Wegelin (2000) for a basic overview. Here we choose to compare against `PLSCanonical` because it treats $X$ and $Y$ symmetrically, and has higher performance than PLS-SVD as it includes the regression step from the score estimates $\hat{v}^z$ onto $X, Y$ to yield $\hat{w}^z$ as the loadings. PLS-SVD directly uses the singular vectors of $XY^T$ as estimates of $\hat{w}^z$, which performs slightly worse, see Figure 4.

For the model with all Gaussian priors, Figure 2a, we find a continuous phase transition between a tractable (easy) regime and an impossible regime. This qualitative phenomenology is the same as in the single-view case (Rangan & Fletcher, 2012; Lesieur et al., 2017). Here the algorithmic threshold obtained from Equation (27) coincides with the Bayes-optimal or information theoretic threshold, $\theta_{\text{IT}} = \theta_{\text{alg}} \approx 1.07$, for $\sqrt{c_v} = 0.75$ and otherwise unit parameters. The weak recovery threshold of the rank-1 spike in each of the views $X, Y$ in isolation ($c_v = 0$) would be $\theta_{\text{IT}}^{\text{single}} = 1$. Therefore, a Bayes-optimal combination of information from the two modalities yields an improvement of the threshold from $\sigma_\xi = 1$ to $\sigma_\xi \approx 1.07$. This improvement grows with the correlation up to $\theta_{\text{IT}} \approx 1.19$ at $c_v = 1$.

There are three observations about the linear methods, as expected from the discussion in Section 2.2 and Section 1.2. Firstly, linearized AMP shares the Bayes-optimal recovery threshold of AMP, but shows sub-optimal performance in estimating $v^z$ when the signal is strong (small $\sigma_\xi$), shown in Figures S3 and S4. Secondly, Figure 3 shows that the performance of CCA is considerably worse than that of PLS even in the presence of large correlation, and away from the regime $\alpha^z < 1$ where the inverse correlation matrices in $(XX^T)^{-1}XY^T(YY^T)^{-1}$ as used by CCA are ill-defined without regularization. Varying the number of samples per feature dimension, $\alpha^z$ in Figure 3b, CCA has highly sub-optimal sample efficiency with $\theta_{\text{CCA}} \approx 3.78$ compared to $\theta_{\text{IT}} \approx 0.04$. Thirdly, PLS gives close to optimal performance in Figure 2 and Figure 3, only its recovery threshold is slightly lower. This difference exacerbates when the correlation between the latent structures decreases, as demonstrated in Figure 4 for $\sqrt{c_v} = 0.2$. As a consequence, while PLS is a practically useful method to extract only the correlated structure of two data views or to predict $Y$ from $X$ in situations with small noise and strongly correlated signals, it is not well-suited for situations with low signal-to-noise ratio: In these cases, even just recovering the low-rank structures using PCA on the individual modalities first and then performing an analysis of the correlation would yield better performance. Of course, the best performance is obtained by combining information of both modalities based on prior information to exploit latent correlations, as done by AMP.

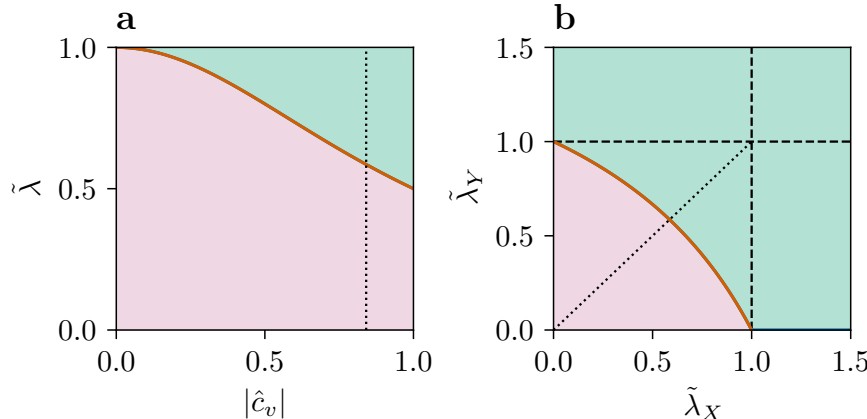

Figure 5: Phase diagram of the algorithmic weak recovery threshold based on (27) and (28). **a** Varying the correlation coefficient for symmetric snr's $\tilde{\lambda}_X = \tilde{\lambda}_Y = \tilde{\lambda}$. **b** The $\tilde{\lambda}_X, \tilde{\lambda}_Y$ plane for $\hat{c}_v^4 = 1/2$. Dashed black lines show the thresholds of the modalities in isolation. The dotted line in **a** and **b** indicates the intersection of both planes.

For the sparse model with Rademacher-Bernoulli prior on $w^z$ with sparsity $\rho_z = 0.02$ in Figure 2**b**, a first-order phase transition is observed instead. Again this qualitative phenomenology matches that of the single-view case (Lesieur et al., 2017), where also a technical discussion of a small regime where the lower branch acquires non-zero overlap is given. Here the algorithmic weak recovery threshold $\theta_{\text{alg}} \approx 0.61$ does not coincide with the IT threshold $\theta_{\text{IT}} \approx 0.71$, instead there is an algorithmically hard phase (Zdeborová & Krzakala, 2016) for $\theta_{\text{alg}} < \sigma_\xi < \theta_{\text{IT}}$ preceding the impossible regime $\sigma_\xi > \theta_{\text{IT}}$. Again some advantage over the single-view threshold $\theta_{\text{alg}}^{\text{single}} \approx 0.57$ is obtained. Note that for tracing the upper branch of the phase diagram with informed AMP, we initialize the iteration at the ground-truth signal.

Finally, in Figure 5 we plot phase diagrams illustrating the algorithmic weak recovery threshold $\theta_{\text{alg}}$ in the reduced three dimensional parameter space of effective snr's $\tilde{\lambda}_X, \tilde{\lambda}_Y$ and correlation coefficient $\hat{c}_v$. Note that $\theta_{\text{IT}}$ may vary depending on the prior and is not shown, while $\theta_{\text{alg}}$ is given by (27) for any zero-mean prior. Figure 5**a** shows for $\tilde{\lambda}_X = \tilde{\lambda}_Y$ the interpolation between zero correlation, equivalent to two single-view models, and perfect correlation, equivalent to the stackable model in Section 2.3. Figure 5**b** illustrates the improvement of the multi-modal threshold over the thresholds of the two isolated single-view models (dashed black lines). Due to the definition of the snr (26), this plot can for example be interpreted as independently varying the aspect ratios $\alpha_z$. Apart from the gain in the lower left sector where no recovery is possible in any isolated model, note that also in the lower-right and upper-left sectors some degree of recovery is always possible in both modalities when the correlation is nonzero, see also Figure S2.

## 4    Conclusions

In order to study the basic properties of multi-modal or multi-view learning, we analysed the Bayes-optimal performance of a correlated matrix factorization problem. Inferring the rank-1 spikes of the matrices corresponds to unsupervised learning of the latent variables underlying the data structure. Allowing for differences in the prior and noise channels across the two modalities or views is shown to alter the combination strategy of the AMP iteration by changing the denoising functions (11) and the $S_z, R_z$ score matrices of the data. Given the combined data, the phenomenology we have observed for the Bayes-optimal learning is qualitatively the same as that of single-view learning, i.e. we have not found additional phase transitions beyond those in the single-view case.

The comparison of the Bayes-optimal weak recovery threshold and those obtained by canonical spectral methods such as PLS and CCA reveals that the canonical methods are suboptimal. This is different from the single-view case, where the optimal algorithmic weak recovery threshold agrees with the threshold present for

the canonical spectral method based on principal component analysis. This difference was not anticipated by the authors nor, to our knowledge, noted in the previous literature, and it is thus worth further investigation.

In future work, it would be interesting to consider a larger number of modalities with a graph of latent relations, as in the original work of Wold (1983) and in structural equation models (Bollen, 1989). Furthermore, natural directions to explore are a supervised version of the task and how neural network-based techniques of multi-modal learning (Baltrušaitis et al., 2018) combine information from the modalities compared to the Bayes-optimal method. Both can readily be approached by considering linear or deep linear methods. An enticing question is how to share information across modalities in an approximately optimal fashion in hierarchical, non-linear models. Clues to this may well be yielded by the ongoing study of multi-sensory integration (Stein et al., 2020) in neuroscience.

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

# A  Relaxed belief propagation

We start from the factor graph Figure 1 and the BP equations (6)-(9). Note the ordering of indices, here we use index $j$ for latent variables and $i$ for $w^z$ variables. The decision to treat the latent variables as one joint variable for the BP messages makes it possible to take into account an arbitrary joint distribution, without splitting $v^X, v^Y$ into shared and independent components - which would yield a rank-2 model with additional messages to keep track of.

First we check that the peculiarity of the double product and joint prior in (8) does not cause additional correlations between the messages $\tilde{n}^X_{kj \to j}(v_j)$ and $\tilde{n}^Y_{kj \to j}(v_j)$ to verify that BP is applicable to this graph. This is not the case because $w^X_i$ and $w^Y_i$ are independent, so conditioned on $(v^X_j, v^Y_j)$, the factors $P^X_{\text{out}}(X_{ij}|w^X_i v^X_j)$ and $P^Y_{\text{out}}(Y_{ij}|w^Y_i v^Y_j)$ in (9) are not correlated; the only dependence could be inherited from $\tilde{m}^X_{ij \to i}(w^X_i)$, $\tilde{m}^Y_{ij \to i}(w^Y_i)$ which appear in $m^z_{i \to ij}(w^z_i)$ both depending on $v^z_j$; however for these the structure of the factor graph is the standard dense type as, e.g. in Lesieur et al. (2017), and the dependence is sufficiently weak given a $\frac{1}{\sqrt{n}}$ scaling of the interactions. Thus (8) does not lead to additional correlations between messages that would compromise the accuracy of the BP iteration.

For convenience we re-state here the channel expansion (5) together with an expansion outside the exponential which will also be used throughout the derivation. Recalling the definitions $S^z_{ij} = \partial_a g_z(z_{ij}, a)|_{a=0}$ and $R^z_{ij} = (\partial_a g_z(z_{ij}, a)|_{a=0})^2 + \partial^2_a g_z(z_{ij}, a)|_{a=0}$, the channels can be expanded either inside or outside the exponent as

$$e^{g_z(z_{ij}, w^z_i v^z_j)} = e^{g_z(z_{ij}, 0) + S^z_{ij}\lambda_z \frac{w^z_i v^z_j}{\sqrt{n_z}} + \frac{1}{2}(R^z_{ij} - (S^z_{ij})^2)\lambda^2_z \frac{(w^z_i v^z_j)^2}{n_z} + \mathcal{O}\left(n_z^{-\frac{3}{2}}\right)}, \tag{A.1}$$

$$= e^{g_z(z_{ij}, 0)}\left[1 + S^z_{ij}\lambda_z \frac{w^z_i v^z_j}{\sqrt{n_z}} + \frac{1}{2}R^z_{ij}\lambda^2_z \frac{(w^z_i v^z_j)^2}{n_z} + \mathcal{O}\left(n_z^{-\frac{3}{2}}\right)\right]. \tag{A.2}$$

In the Gaussian noise case, $S^z_{ij} = \frac{z_{ij}}{\sigma^2_{\xi z}}$ and $R^z_{ij} = \frac{z^2_{ij}}{\sigma^4_{\xi z}} - \frac{1}{\sigma^2_{\xi z}}$.

Now to obtain rBP we use that the BP equations close on the Gaussian statistics of the messages, leading to an iteration on the means and variances of the beliefs. Plugging (A.2) into (7) (and analogously (9)) we get at the example of $\tilde{m}^X_{ij \to i}$

$$\tilde{m}^X_{ij \to i}(w_i) = \frac{e^{g(X_{ij}, 0)}}{\mathcal{Z}^{X,m}_{ij \to i}} \int dv^X_j dv^Y_j \, n^X_{j \to ij}(v^X_j, v^Y_j)\left[1 + S^X_{ij}\lambda_X \frac{w^X_i v^X_j}{\sqrt{n_X}} + \frac{1}{2}R^X_{ij}\lambda^2_X \frac{(w^X_i v^X_j)^2}{n_X} + \mathcal{O}\left(n_X^{-\frac{3}{2}}\right)\right], \tag{A.3}$$

which is clearly a function of the mean and variance (the covariance $\text{Cov}[v^X_j v^Y_j]$ does not appear, since in $\tilde{m}^z_{ij \to i}$ only the marginalized $\int dv^z_j n^z_{j \to ij}(v^X_j, v^Y_j)$ are present)

$$\hat{v}^X_{j \to ij} = \int dv^X_j dv^Y_j \, n^X_{j \to ij}(v^X_j, v^Y_j)v_j \tag{A.4}$$

$$\hat{\sigma}^X_{v, j \to ij} = \int dv^X_j dv^Y_j \, n^X_{j \to ij}(v^X_j, v^Y_j)v^2_j - (\hat{v}^X_{j \to ij})^2, \tag{A.5}$$

so that

$$\tilde{m}^X_{ij \to i}(w^X_i) = \frac{1}{\mathcal{Z}^{X,m}_{ij \to i}} \exp\left[g(X_{ij}, 0) + S^X_{ij}\lambda_X \frac{w^X_i \hat{v}^X_{j \to ij}}{\sqrt{n_X}} - \frac{1}{2}(S^X_{ij})^2\lambda^2_X \frac{(w^X_i)^2(\hat{v}^X_{j \to ij})^2}{n_X}\right. \tag{A.6}$$

$$\left. + \frac{1}{2}R^X_{ij}\lambda^2_X \frac{(w^X_i)^2((\hat{v}^X_{j \to ij})^2 + \hat{\sigma}^X_{v, j \to ij})}{n_X} + \mathcal{O}\left(n_X^{-\frac{3}{2}}\right)\right], \tag{A.7}$$

where we exploited the exponential form of the expansion, (5). Plugging this into (6) , and doing the analogous steps for (8), we find

$$m_{i \to ij}^z(w_i^z) = \frac{P_{w^z}(w_i^z)}{\mathcal{Z}_{i \to ij}^{z,m}} \exp\left( J_{z,i \to ij}^w w_i^z - \frac{1}{2} K_{z,i \to ij}^w (w_i^z)^2 \right) \tag{A.8}$$

$$n_{j \to ij}^z(v_j^X, v_j^Y) = \frac{P_v(v_j^X, v_j^Y)}{\mathcal{Z}_{j \to ij}^{z,n}} \exp\left( J_{z,j \to ij}^v v_j^z + J_{\bar{z},j}^v v_j^{\bar{z}} - \frac{1}{2} K_{z,j \to ij}^v (v_j^z)^2 - \frac{1}{2} K_{\bar{z},j}^v )(v_j^{\bar{z}})^2 \right) \tag{A.9}$$

where the factors $e^{g_z(z_{ij},0)}$ have been absorbed in the normalization, we see that the form of the message distribution is of the tilted prior type $\mathcal{W}(x, K, J) = P_x(x) \exp(Jx - \frac{1}{2} x^T K x)$ (10), with the source terms $J_{z,i \to ij}^w$, $K_{z,i \to ij}^w$ and $J_{z,j \to ij}^v$, $K_{z,j \to ij}^v$ given by

$$J_{z,j \to ij}^{v,t} = \frac{\lambda_z}{\sqrt{n_z}} \sum_{k \neq i}^{n_z} S_{kj}^z \hat{w}_{k \to kj}^{z,t-1} \tag{A.10}$$

$$K_{z,j \to ij}^{v,t} = \frac{\lambda_z^2}{n_z} \sum_{k \neq i}^{n_z} \left[ (S_{kj}^z \hat{w}_{k \to kj}^{z,t-1})^2 - R_{kj}^z ((\hat{w}_{k \to kj}^{z,t-1})^2 + \hat{\sigma}_{w,k \to kj}^{z,t-1}) \right], \tag{A.11}$$

$$J_{z,i \to ij}^{w,t} = \frac{\lambda_z}{\sqrt{n_z}} \sum_{k \neq j}^{d} S_{ik}^z \hat{v}_{k \to ik}^{z,t} \tag{A.12}$$

$$K_{z,i \to ij}^{w,t} = \frac{\lambda_z^2}{n_z} \sum_{k \neq j}^{d} \left[ (S_{ik}^z \hat{v}_{k \to ik}^{z,t})^2 - R_{ik}^z ((\hat{v}_{k \to ik}^{z,t})^2 + \hat{\sigma}_{v,k \to ik}^{z,t}) \right] \tag{A.13}$$

where we added also explicit time indices for updating first $n_{v \to ij}^z(v_j^z)$ then $m_{i \to ij}^z(w_i^z)$, and the notation $J_{\bar{z},j}^v$, $K_{\bar{z},j}^v$ in Equation (A.9) signifies that the term $k = i$ is not excluded from the summation, which eliminates the dependence on the target node $i$. Lastly, $\hat{w}_{k \to kj}^{z,t}$ and $\hat{\sigma}_{w,k \to kj}^{z,t}$ are defined as mean and variance analogous to (A.4,A.5), but of the messages $m_{i \to ij}^z(w_i^z)$. With the above source terms and the definition of the denoising function (11) as the first derivative of the cumulant generating function, the rBP equations with sequential update order (first $v$, then $w$) are thus

$$\hat{v}_{j \to ij}^{z,t} = f_{\text{in}}^{v^z}(K_{X,j \to ij}^{v,t}, K_{Y,j \to ij}^{v,t}, J_{X,j \to ij}^{v,t}, J_{Y,j \to ij}^{v,t}) \tag{A.14}$$

$$\hat{\sigma}_{v,j \to ij}^{z,t} = \frac{\partial f_{\text{in}}^{v^z}}{\partial J_z}(K_{X,j \to ij}^{v,t}, K_{Y,j \to ij}^{v,t}, J_{X,j \to ij}^{v,t}, J_{Y,j \to ij}^{v,t}). \tag{A.15}$$

$$\hat{w}_{i \to ij}^{z,t} = f_{\text{in}}^w(K_{z,i \to ij}^{w,t}, J_{z,i \to ij}^{w,t}) \tag{A.16}$$

$$\hat{\sigma}_{w,i \to ij}^{z,t} = \frac{\partial f_{\text{in}}^w}{\partial J_z}(K_{z,i \to ij}^{w,t}, J_{z,i \to ij}^{w,t}) \tag{A.17}$$

While taking it into account in the following derivations, for ease of notation we have omitted in Equations (A.14) and (A.15) above the fact that, as explicit in Equation (A.9), for $\hat{v}_{j \to ij}^{z,t+1}$ the respective $\bar{z}$ source terms do not exclude the index $i$ in the sum, $K_{\bar{z},j \to ij}^{v,t} \to K_{\bar{z},j}^{v,t}$ and $J_{\bar{z},j \to ij}^{v,t} \to J_{\bar{z},j}^{v,t}$.

## B  AMP: closing on the marginals

The rBP iteration works with $\mathcal{O}(d^2)$ truncated marginals on the edges of the factor graph, but can be approximated by an AMP iteration operating on the $d + n_X + n_Y$ full marginals of the nodes. This is possible since $J_{z,i \to ij}^{w,t}$, $K_{z,i \to ij}^{w,t}$ and $J_{z,j \to ij}^{v,t}$, $K_{z,j \to ij}^{v,t}$ depend only weakly on the target factor node. However one can not naively neglect the dependence of $\hat{v}_{j \to ij}^{z,t}$ and $\hat{w}_{i \to ij}^{z,t}$ on the target node, since a consistent expansion in $\mathcal{O}(n_z^{-\frac{1}{2}})$ results in the $\mathcal{O}(1)$ Onsager reaction terms which need to be taken into account in the estimators of the marginals' means.

First, consider $J_{z,i}^{w,t}$, $K_{z,i}^{w,t}$ and $J_{z,j}^{v,t}$, $K_{z,j}^{v,t}$ which we get by not excluding the $k = j$ term (or $k = i$ respectively) from the summation in (A.12-A.11). For example,

$$J_{z,i}^{w,t} = \frac{\lambda_z}{\sqrt{n_z}} \sum_k^d S_{ik}^z \hat{v}_{k \to ik}^{z,t}. \tag{B.1}$$

Due to the prefactor, by adding one term to the sum we make the errors

$$J_{z,i \to ij}^{w,t} - J_{z,i}^{w,t} = -\frac{\lambda_z}{\sqrt{n_z}} S_{ij}^z \hat{v}_{j \to ij}^{z,t} = \mathcal{O}(n_z^{-\frac{1}{2}}) \tag{B.2}$$

$$J_{z,j \to ij}^{v,t} - J_{z,j}^{v,t} = -\frac{\lambda_z}{\sqrt{n_z}} S_{ij}^z \hat{w}_{i \to ij}^{z,t-1} = \mathcal{O}(n_z^{-\frac{1}{2}}) \tag{B.3}$$

and $\mathcal{O}(n_z^{-1})$ in the cases of $K_{z,i}^{w,t}, K_{z,j}^{v,t}$, all negligible at $n_z \gg 1$. Note that here we also assumed that at each time-step, the $\hat{v}^t$ are updated first based on $\hat{w}^{t-1}$, and then the $\hat{w}^t$ based on $\hat{v}^t$, as discussed in Section 2.1. Next we want to replace also the means by target-independent versions $\hat{v}_j^{z,t}$ and $\hat{w}_i^{z,t}$ and so the variances by $\hat{\sigma}_{v,j}^{z,t}$ and $\hat{\sigma}_{w,j}^{z,t}$. The errors we make with this replacement,

$$\hat{w}_{i \to ij}^{z,t} - \hat{w}_i^{z,t} = f_{\text{in}}^w(K_{z,i \to ij}^{w,t}, J_{z,i \to ij}^{w,t}) - f_{\text{in}}^w(K_{z,i}^{w,t}, J_{z,i}^{w,t}) \tag{B.4}$$

$$= -\frac{\lambda_z}{\sqrt{n_z}} S_{ij}^z \hat{\sigma}_{w,i \to ij}^{z,t} \hat{v}_{j \to ij}^{z,t} + \mathcal{O}(\frac{1}{n_z}) \tag{B.5}$$

$$\hat{v}_{j \to ij}^{z,t} - \hat{v}_j^{z,t} = -\frac{\lambda_z}{\sqrt{n_z}} S_{ij}^z \hat{\sigma}_{v,j \to ij}^{z,t} \hat{w}_{i \to ij}^{z,t-1} + \mathcal{O}(\frac{1}{n_z}), \tag{B.6}$$

are relevant since plugging into (B.1) we get errors $\sim \frac{1}{n_z}(S_{ik}^z)^2$ which have non-vanishing mean of $\mathcal{O}(\frac{1}{n_z})$; thus replacing each of the $d$ terms of the sum in $J_{z,i}^{w,t}$, and $n_z$ terms of the sum in $J_{z,j}^{v,t}$, results in a compound error of $\mathcal{O}(\frac{d}{n_z})$ and $\mathcal{O}(1)$ respectively. Therefore, the Onsager correction terms (B.5) and (B.6) need to be added to the linear source terms $J_{z,i}^{w,t}$ and $J_{z,j}^{v,t}$ of the AMP iteration, yielding Equations (12) to (15) in the main text.

## B.1 Denoising functions for Gaussian priors

For multivariate Gaussian prior $P_x \sim \mathcal{N}(0, \Sigma)$, completing the square in the resulting product of Gaussians in (10) (with diagonal quadratic source terms, so $K$ is a vector),

$$\mathcal{W}(x, K, J) = \frac{1}{2\pi\sqrt{\det \Sigma}} \exp\left(-\frac{1}{2}x^T \Sigma^{-1} x + J^T x - \frac{1}{2}x^T \text{diag}(K)x\right) \tag{B.7}$$

$$= \frac{1}{2\pi\sqrt{\det \Sigma}} \exp\left(-\frac{1}{2}\left(x - \tilde{\Sigma}_K J\right)^T \tilde{\Sigma}_K^{-1}\left(x - \tilde{\Sigma}_K J\right) + \frac{1}{2}J^T \tilde{\Sigma}_K J\right) \tag{B.8}$$

where $\tilde{\Sigma}_K = (\Sigma^{-1} + \text{diag}(K))^{-1}$. To obtain $f_{\text{in}}^{v^z}(K, J) = \partial_{J_{1/2}} \log \int \text{d}x \, \mathcal{W}^v(x, K, J)$ in the two-dimensional case, we use that the mean of a distribution is equal to the mean of its marginals, such that

$$f_{\text{in}}^{v^z}(K_1, K_2, J_1, J_2) = (\tilde{\Sigma}_K J)_z = \tilde{\Sigma}_{zz}(K)J_z + \tilde{\Sigma}_{z\bar{z}}(K)J_{\bar{z}}. \tag{B.9}$$

Twice applying $2 \times 2$ matrix inversion, the components of $\tilde{\Sigma}_K$ are given by

$$\begin{pmatrix} \tilde{\Sigma}_{XX}(K) & \tilde{\Sigma}_{XY}(K) \\ \tilde{\Sigma}_{YX}(K) & \tilde{\Sigma}_{YY}(K) \end{pmatrix} = \frac{\det \Sigma}{(\sigma_{v^X}^2 + K_2 \det \Sigma)(\sigma_{v^Y}^2 + K_1 \det \Sigma) - c_v^2} \begin{pmatrix} \sigma_{v^X}^2 + K_2 \det \Sigma & c_v \\ c_v & \sigma_{v^Y}^2 + K_1 \det \Sigma \end{pmatrix}, \tag{B.10}$$

with $\det \Sigma = \sigma_{v^X}^2 \sigma_{v^Y}^2 - c_v^2$.
For the scalar Gaussian prior $P_w^z \sim \mathcal{N}(0, \sigma_{w^z}^2)$, the result is simply

$$f_{\text{in}}^{w^z}(K, J) = \frac{J}{K + \sigma_{w^z}^{-2}}. \tag{B.11}$$

## B.2 Denoising function for Rademacher-Bernoulli prior

We consider $w^z$ sparse with Rademacher-Bernoulli prior $P_{w^z}^{RB}(w_i^z)\frac{\rho_z}{2}[\delta(w_i^z-1)+\delta(w_i^z+1)]+(1-\rho_z)\delta(w_i^z)$. For small $\rho_z$ a hard phase due to a first-order transition is expected, while for $\rho_z \to 1$ the upper branch deforms until a continuous transition is recovered.

Now for $P^{RB}$ the cumulant generating function of the tilted prior distribution (10) becomes

$$\log \mathcal{Z}_{w^z}(K,J) = \log \int \mathrm{d}w^z \, P_{w^z}^{RB}(w^z) \exp\left(Jw^z - \frac{1}{2}K(w^z)^2\right)$$

$$= \log\left[\rho_z \cosh(J)e^{-\frac{1}{2}K} + (1-\rho_z)\right] \tag{B.12}$$

so that the mean or the denoising function is

$$f_{\mathrm{in}}^{w^z}(K,J) = \frac{\partial}{\partial J}\log \mathcal{Z}_{w^z}(K,J)$$

$$= \frac{\rho_z \sinh(J)e^{-\frac{1}{2}K}}{\rho_z \cosh(J)e^{-\frac{1}{2}K} + (1-\rho_z)}. \tag{B.13}$$

$$= \frac{\rho_z \tanh(J)}{\rho_z + \frac{2(1-\rho_z)}{\exp(J-\frac{1}{2}K)+\exp(-J-\frac{1}{2}K)}} \tag{B.14}$$

where the last version (B.14) is stable against floating point overflows in `numpy`, that is it avoids any `np.nan` by avoiding `0*np.inf` or `0/0` or `np.inf/np.inf` to occur, and used in the numerical implementations. For the derivative, a numerically benign version is

$$\frac{\partial f_{\mathrm{in}}^{w^z}}{\partial J}(K,J) = \frac{\rho_z^2}{((1-\rho_z)e^{\frac{1}{2}K}+\rho_z \cosh(J))^2} + \frac{\rho_z(1-\rho_z)}{(1-\rho_z+\rho_z r(K,J))^2}$$

$$\frac{\rho_z}{\left(\frac{1-\rho_z}{r(K,J)}+2\rho_z(1-\rho_z)+\rho_z^2 r(K,J)\right)} - \frac{\rho_z}{1-\rho_z+\rho_z r(K,J)} \tag{B.15}$$

with $r(K,J) = \frac{1}{2}e^{J-\frac{1}{2}K} + \frac{1}{2}e^{-J-\frac{1}{2}K}$.

## B.3 Initialization of AMP

For sparse priors, AMP is known to have convergence problems for small noise at finite size, and when the trajectory leaves the proximity of the Nishimori line. Drift away from the Nishimori line arises in particular due to finite size noise close to the first-order transition. While also caused by additional factors such as nonzero mean of the data (Caltagirone et al., 2014) and there exist principled (Vila et al., 2015; Rangan et al., 2019) and non-principled (Sterk et al., 2023) mitigation techniques, these issues are importantly caused and partly avoidable by the initialization method.

Note that the initialization requires not only to choose the mean estimators $\hat{w}^{z,t_0}$, but also the variance estimators $\hat{\sigma}_{w/v}^{z,t_0}$ and the value $\hat{v}^{z,t_0}$ from the past time step for the Onsager correction terms. We choose the variances as those of the prior and the past time step value $\hat{v}^{z,t_0}$ as zero in both versions below.

For small noise at finite size, that is $\sigma_\xi^2\sqrt{n}\sim\mathcal{O}(1)$, the expansion in $n^{-1/2}$ made in the derivation of rBP and AMP looses its accuracy. Here the well known spectral initialization is beneficial. It leaves the Nishimori line, but results in reliable convergence if the signal is strong (Celentano et al., 2023).

For moderate or larger noise, the average distance of the initialization from the Nishimori line can be minimized by rescaling a random sample from the prior such that the relation $Q_{w/v}^z = |M_{w/v}^z|$ holds on expectation for the given finite system size. This yields $\sigma_{\mathrm{init}}^2 = \frac{\sigma_{\mathrm{prior}}^2}{n}$ for a vector in $\mathbb{R}^n$. Note that the distribution of the random overlap is still centered on zero, so this initialization can only minimize the average distance from the Nishimori line, not eliminate it, therefore we refer to it as "approximate Nishimori". To enforce the condition on the level of the single realization would require information about the ground-truth direction to enter the algorithm.

## C   Linearized AMP: optimal spectral algorithm for weak recovery

---

**Algorithm 2** linearized AMP

---

**Input:**

data $X, Y$

parameters $\lambda_z, \sigma_{\xi^z}, \Sigma_v, \sigma_{w^z}$ for $z \in \{X, Y\}$

**Initialize:**

# random guess from prior (iid. normal also works)

sample $\hat{w}^z \sim P_{w^z}$ and $\hat{v}^z \sim P_{v^z}$

$\hat{w} \leftarrow \left(\hat{w}^X, \hat{w}^Y\right)^T$

$\hat{v} \leftarrow \left(\hat{v}^X, \hat{v}^Y\right)^T$

$\Gamma_v, \Gamma_w \leftarrow$ Equations (17) and (18)

**Run:** # Power iteration

**while not converged do**

    $\hat{w} \leftarrow \Gamma_w \hat{w}$

    $\hat{v} \leftarrow \Gamma_v \hat{v}$

    $\hat{v} \leftarrow \frac{\hat{v}}{||\hat{v}||}$

    $\hat{w} \leftarrow \frac{\hat{w}}{||\hat{w}||}$

**end while**

# (optionally scale norms to expected norm of the prior)

**return** $\hat{w}^z, \hat{v}^z$ for $z \in \{X, Y\}$

---

For priors of mean zero, we expand the rBP equations (A.14)-(A.17) around $\hat{w}^z_{i \to ij}, \hat{v}^z_{j \to ij} = 0$ to obtain a linearized rBP iteration, which is nothing but a power iteration of a linear operator. Again the the dimension of the operator can be reduced to $2d \times (n_x + n_y)$ by the analogous steps as in Appendix B to obtain a power-iteration on the node level.

First, we use that the $\partial_{v^z}$ and $\partial_{w^z}$ derivatives of both $\hat{\sigma}^z_{v,j \to ij}$ and $\hat{\sigma}^z_{w,j \to ij}$ with respect to both $\hat{w}^z_{i \to ij}$ and $\hat{v}^z_{j \to ij}$ are zero at the origin; this follows from the $\mathbb{Z}^2$ symmetry of choosing the sign of the estimated vectors (only the relative sign between $\hat{v}^X_{j \to ij}$ and $\hat{v}^Y_{j \to ij}$ matters). Consistently with this argument, seeing that $\hat{w}^z_{i \to ij}$ and $\hat{v}^z_{j \to ij}$ appear squared in $K^{w,t}_{z,i \to ij}$ and $K^{v,t}_{z,j \to ij}$, their derivatives at the origin are vanishing as well. We are then left with computing

$$\left.\frac{\partial \hat{v}^{z,t}_{j \to ij}}{\partial \hat{w}^{z,t-1}_{k \to kj}}\right|_{w=0} = \left.\frac{\partial f^{v^z}_{\text{in}}}{\partial J^z}\right|_{w=0} \left.\frac{\partial J^{v,t-1}_{z,j \to ij}}{\partial \hat{w}^{z,t-1}_{k \to kj}}\right|_{w=0} = \sigma^2_{v^z} \frac{\lambda_z}{\sqrt{n_z}} S^z_{kj} \qquad \forall k \neq i \tag{C.1}$$

$$\left.\frac{\partial \hat{v}^{z,t}_{j \to ij}}{\partial \hat{w}^{\bar{z},t-1}_{k \to kj}}\right|_{w=0} = \left.\frac{\partial f^{v^z}_{\text{in}}}{\partial J^{\bar{z}}}\right|_{w=0} \left.\frac{\partial J^{v,t-1}_{\bar{z},j \to ij}}{\partial \hat{w}^{\bar{z},t-1}_{k \to kj}}\right|_{w=0} = c_v \frac{\lambda_{\bar{z}}}{\sqrt{n_{\bar{z}}}} S^{\bar{z}}_{kj} \qquad \forall k \tag{C.2}$$

$$\left.\frac{\partial \hat{w}^{z,t}_{i \to ij}}{\partial \hat{v}^{z,t}_{k \to ik}}\right|_{v=0} = \left.\frac{\partial f^{w^z}_{\text{in}}}{\partial J^z}\right|_{v=0} \left.\frac{\partial J^{w,t}_{z,i \to ij}}{\partial \hat{v}^{z,t}_{k \to ik}}\right|_{v=0} = \sigma^2_{w^z} \frac{\lambda_z}{\sqrt{n_z}} S^z_{ik} \qquad \forall k \neq j \tag{C.3}$$

where we have used that $f_{\text{in}}$ is defined as a derivative of the cumulant generating function of $\mathcal{W}$, so the derivatives evaluated at zero give the prior (co)variances. Note the flip of $z \to \bar{z}$ between the first and the second line.

In the first and the second line we had to exclude the $k = i$ and $k = j$ index, respectively, where the derivative would be zero. Apart from this, the derivatives are completely independent of the target node of the messages. In analogy to the derivation of AMP, the error made by adding these two terms in order to get an iteration

on the node level is

$$\hat{w}_i^{z,t} - \hat{w}_{i\to ij}^{z,t} = \frac{\lambda_z^2}{n_z}\sigma_{v^z}^2\sigma_{w^z}^2\left(\sum_{k\neq j}S_{ik}^z S_{ik}^z \hat{w}_{i\to ij}^{z,t-1} + S_{ij}^z\sum_k S_{kj}^z\hat{w}_{k\to kj}^{z,t-1}\right) \tag{C.4}$$

$$= \frac{\lambda_z^2}{n_z}\sigma_{v^z}^2\sigma_{w^z}^2\left(\hat{w}_{i\to ij}^{z,t-1}\sum_k (S_{ik}^z)^2 + \mathcal{O}(\frac{1}{\sqrt{n_z}})\right) \tag{C.5}$$

$$\hat{v}_j^{z,t} - \hat{v}_{j\to ij}^{z,t} = \frac{\lambda_z^2}{n_z}\sigma_{v^z}^2\sigma_{w^z}^2\left(\hat{v}_{j\to ij}^{z,t-1}\sum_k (S_{kj}^z)^2 + \mathcal{O}(\frac{1}{\sqrt{n_z}})\right) \tag{C.6}$$

$$\tag{C.7}$$

where we are directly considering the products of the operators updating $\hat{v}^z$ and $\hat{w}^z$, to get two iterations running only on the $\hat{v}^z$ and $\hat{w}^z$ vectors, respectively. The Onsager reactions $\sum_k (S_{ik}^z)^2 \sim \mathcal{O}(1)$ and $\sum_k (S_{kj}^z)^2 \sim \mathcal{O}(1)$ can not be neglected (we add the $k=j$ and $k=i$ terms here since they are sub-leading). Therefore going from linearized rBP to linearized AMP, we find that the Onsager correction is exactly to subtract the terms on the diagonal of the matrix, giving the block structured matrices $\Gamma_v$ and $\Gamma_w$ in Equations (17) and (18). Note that directly linearizing the AMP equations would make it necessary to show that the dependence of the linear $\hat{v}^z$ iteration on the Onsager reaction of the intermediate $\hat{w}^z$ update step is vanishing, and vice versa for the linear $\hat{w}^z$ iteration. A simple way to see this is by starting from linearizing rBP.

## D  State evolution

By introducing a set of $\mathcal{O}(1)$ order parameters we now find low-dimensional effective equations which describe the rBP dynamics in the thermodynamic limit. Note that one would like to get the dynamics of the overlaps between the full marginal estimates (the messages where the target index in the sum is not excluded) and the signal. While the rBP iteration runs on the truncated marginals with excluded target index, the difference in the thermodynamic limit is vanishing, $\langle \hat{w}_i w_i^0\rangle - \langle \hat{w}_{i\to ij} w_i^0\rangle \sim \mathcal{O}(\frac{1}{\sqrt{n}})$ , and we can replace the overlaps of the full marginals by those of the truncated marginals. So we introduce the order parameters

$$M_w^{z,t} = \frac{1}{n_z - 1}\sum_{i\neq j}^{n_z}\hat{w}_{i\to ij}^{z,t}w_{z,i}^0 \qquad\qquad M_v^{z,t} = \frac{1}{d-1}\sum_{j\neq i}^d \hat{v}_{i\to ij}^{z,t}v_{z,j}^0 \tag{D.1}$$

$$Q_w^{z,t} = \frac{1}{n_z - 1}\sum_{i\neq j}^{n_z}\hat{w}_{i\to ij}^{z,t}\hat{w}_{i\to ij}^{z,t} \qquad\qquad Q_v^{z,t} = \frac{1}{d-1}\sum_{j\neq i}^d \hat{v}_{i\to ij}^{z,t}\hat{v}_{i\to ij}^{z,t} \tag{D.2}$$

$$\Sigma_w^{z,t} = \frac{1}{n_z - 1}\sum_{i\neq j}^{n_z}\hat{\sigma}_{w,i\to ij}^{z,t} \qquad\qquad \Sigma_v^{z,t} = \frac{1}{d-1}\sum_{j\neq i}^d \hat{\sigma}_{v,i\to ij}^{z,t} \tag{D.3}$$

where $w_z^0, v_z^0$ are the ground-truth factors. Notice that we drop the $j$ index for the order parameters, because in the thermodynamic limit they all concentrate and become independent of j.

In the following, we exploit self-averaging in several places; any node-averaged quantity concentrates to its mean over noise disorder, which also allows us to drop indices for iid. quantities. Given a quantity $f_{kl} \sim$ iid. (or with weak enough correlations) and with $\text{Var}(f_{kl}) = \sigma_f^2 \sim \mathcal{O}(1)$ and $\mathbb{E}(f_{kl}) = f \sim \mathcal{O}(1)$ we have

$$\frac{1}{d}\sum_k^d f_{kl} = f + \mathcal{O}\left(\frac{1}{\sqrt{d}}\right) = \mathbb{E}(f_{kl}) + \mathcal{O}\left(\frac{1}{\sqrt{d}}\right). \tag{D.4}$$

Note that we need to be careful with applying this in case of vanishing mean $f = 0$, since then the leading order term $\sim \mathcal{O}((d)^{-\frac{1}{2}})$ may or may not be negligible, depending on the context.

Since the order parameters are self-averaging we replace the sum over node indices by an average over the disorder, and write their update equations by plugging in the rBP equations (A.14-A.17). At the example of

$M_w$,

$$M_w^{z,t} = \mathbb{E}_{w_z^0, K_{z,i\to ij}^{w,t}, J_{z,i\to ij}^{w,t}}[f_{\text{in}}^w(K_{z,i\to ij}^{w,t}, J_{z,i\to ij}^{w,t})w_z^0]. \tag{D.5}$$

Therefore we need to find the mean and variance of the source terms (14)-(13) which become Gaussian for $d \to \infty$, across noise realizations of the observations $z$.

While not requiring Bayes-optimality, so that the priors and noise channels of ground-truth and algorithm can differ, e.g. $P_{\text{out}}^0(z_{ij}|w_i, v_j) = e^{g_z^0(z_{ij}, w_i^z v_j^z)} \neq e^{g_z(z_{ij}, w_i^z v_j^z)}$, we do assume the following property holds

$$\forall w_i, v_j \quad \int dz_{ij} \, P_{\text{out}}^0(z_{ij}|w_i^z, v_j^z) \frac{\partial g_z(z_{ij}|w_i^z, v_j^z)}{\partial w_i^z / v_j^z} = 0, \tag{D.6}$$

which in the Bayes-optimal case $P_{\text{out}}^0(z_{ij}|w_i^z, v_j^z) = e^{g_z(z_{ij}|w_i^z, v_j^z)}$ follows directly from normalization. For a discussion of when this is satisfied, refer to Lesieur et al. (2017), p.34. We do the mean and variance calculation first at the example of $J_{z,i\to ij}^{w,t}$. The mean is

$$\mathbb{E}(J_{z,i\to ij}^{w,t}) = \frac{\lambda_z}{\sqrt{n_z}} \sum_{k\neq i}^d \int dz_{ik} \, P_{\text{out}}^0(z_{ik}|w_{z,i}^0 \, v_{z,k}^0) S_{ik}^z \hat{v}_{k\to ik}^{z,t} \tag{D.7}$$

$$= \frac{\lambda_z}{\sqrt{n_z}} \sum_{k\neq i}^d \int dz_{ik} \, P_{\text{out}}^0(z_{ik}|0) \left[ 1 + \lambda_z^0 \frac{w_{z,i}^0 \, v_{z,k}^0}{\sqrt{n_z}} S_{ik}^{0,z} + \mathcal{O}(\frac{1}{n_z}) \right] S_{ik}^z \hat{v}_{k\to ik}^{z,t} \tag{D.8}$$

$$= \frac{\lambda_z \lambda_z^0}{\hat{\Delta}^z} w_{z,i}^0 \, \mathbb{E}_{P_{\text{out}}^0(z|0)} \left[ \frac{1}{n_z} \sum_{k\neq i}^d \hat{v}_{k\to ik}^{z,t} v_{z,k}^0 \right] + \mathcal{O}(\frac{1}{\sqrt{n_z}}) \tag{D.9}$$

$$= \frac{\alpha_z \lambda_z \lambda_z^0}{\hat{\Delta}^z} M_v^{z,t} w_{z,i}^0 + \mathcal{O}(\frac{1}{\sqrt{n_z}}). \tag{D.10}$$

where in the second to third line, using that $S_{ik}^{0,z} S_{ik}^z$ and $\hat{v}_{k\to ik}^{z,t} v_{z,k}^0$ are approximately independent both w.r.t. indices and noise realization (note that the integration is over $P_{\text{out}}^0(z_{ik}|0)$), we defined

$$\frac{1}{\hat{\Delta}^z} = \mathbb{E}_{P_{\text{out}}^0(z|0)} \left[ S_{ik}^{0,z} S_{ik}^z \right] \tag{D.11}$$

and in the last line could get rid of the expectation over the channel noise by plugging in the self-averaging order parameter. Next, the variance of $J_{z,i\to ij}^{w,t}$ gives

$$\text{Var}(J_{z,i\to ij}^{w,t}) = \frac{\lambda_z^2}{n_z} \sum_{k,l\neq i}^d \mathbb{E}_{P_{\text{out}}^0(z_{ik}|w_{z,i}^0 \, v_{z,k}^0)} \mathbb{E}_{P_{\text{out}}^0(z_{il}|w_{z,i}^0 \, v_{z,l}^0)} \left[ S_{ik}^z S_{il}^z \hat{v}_{k\to ik}^{z,t} \hat{v}_{l\to il}^{z,t} \right] - \mathbb{E}(J_{z,i\to ij}^{w,t})^2 \tag{D.12}$$

$$= \frac{\lambda_z^2}{n_z} \sum_{k\neq i}^d \mathbb{E}_{P_{\text{out}}^0(z|0)} \left[ \left( S_{ik}^z \hat{v}_{k\to ik}^{z,t} \right)^2 + \mathcal{O}(\frac{1}{\sqrt{n_z}}) \right] + \mathcal{O}(\frac{1}{d}) \tag{D.13}$$

$$= \frac{\lambda_z^2}{\tilde{\Delta}^z} \mathbb{E}_{P_{\text{out}}^0(z|0)} \left[ \frac{1}{n_z} \sum_{k\neq i}^d \hat{v}_{k\to ik}^{z,t} \hat{v}_{k\to ik}^{z,t} \right] + \mathcal{O}(\frac{1}{\sqrt{n_z}}) \tag{D.14}$$

$$= \frac{\alpha_z \lambda_z^2}{\tilde{\Delta}^z} Q_v^t + \mathcal{O}(\frac{1}{\sqrt{n_z}}) \tag{D.15}$$

where in the first line the mean subtraction cancels with the $k \neq l$ terms up to the one term which gives the $\mathcal{O}(\frac{1}{d})$ in the second line, and then we use that, for the remaining diagonal terms the zeroth order in the expansion of $P_{\text{out}}^0(z_{ik}|w_{z,i}^0/u_i^0 \, v_{z,k}^0)$ is already non-vanishing. Then, along the line of the arguments for $\mathbb{E}(J_{z,i\to ij}^{w,t})$, we defined

$$\frac{1}{\tilde{\Delta}^z} = \mathbb{E}_{P_{\text{out}}^0(z|0)} \left[ S_{ik}^z S_{ik}^z \right]. \tag{D.16}$$

For $K_{z,i \to ij}^{w,t}$, each term in (15) is self-averaging, so the variance is sub-leading:

$$\mathbb{E}(K_{z,i \to ij}^{w,t}) = \frac{\alpha_z \lambda_z^2}{\tilde{\Delta}^z} Q_v^{z,t} - \alpha_z \lambda_z^2 \bar{R}^z (Q_v^{z,t} + \Sigma_v^{z,t}) + \mathcal{O}(\frac{1}{\sqrt{n_z}}) \tag{D.17}$$

$$\text{Var}(K_{z,i \to ij}^{w,t}) = \mathcal{O}(\frac{1}{\sqrt{n_z}}), \tag{D.18}$$

where we used approximate independence of $R_{ik}^z$ and $((\hat{v}_{k \to ik}^{z,t})^2 + \hat{\sigma}_{v,k \to ik}^{z,t})$ as before for $S_{ij}^z$ and defined using self-averaging (D.4)

$$\bar{R}^z = \mathbb{E}_{P_{\text{out}}^0(z_{ik})}(R_{ik}^z) = \frac{1}{d} \sum_{k \neq i}^{d} R_{ik}^z + \mathcal{O}(\frac{1}{\sqrt{d}}). \tag{D.19}$$

Analogously,

$$\mathbb{E}(J_{z,j \to ij}^{v,t}) = \frac{\lambda_z \lambda_z^0}{\hat{\Delta}^z} M_w^{z,t-1} v_{z,j}^0 + \mathcal{O}(\frac{1}{\sqrt{n_z}}) \tag{D.20}$$

$$\text{Var}(J_{z,j \to ij}^{v,t}) = \frac{\lambda_z^2}{\tilde{\Delta}^z} Q_w^{z,t-1} + \mathcal{O}(\frac{1}{\sqrt{n_z}}) \tag{D.21}$$

$$\mathbb{E}(K_{z,j \to ij}^{v,t}) = \frac{\lambda_z^2}{\tilde{\Delta}^z} Q_w^{z,t-1} - \lambda_z^2 \bar{R}^z (Q_w^{z,t-1} + \Sigma_w^{z,t-1}) + \mathcal{O}(\frac{1}{\sqrt{n_z}}) \tag{D.22}$$

$$\text{Var}(K_{z,j \to ij}^{v,t}) = \mathcal{O}(\frac{1}{\sqrt{n_z}}). \tag{D.23}$$

Due to the exchange of node and disorder averages the means and variances are independent of the $i, j$ indices, so that we drop them. Also it does not make a difference with the truncation at $\mathcal{O}(\frac{1}{\sqrt{n}})$ whether the $i = j$ term is included in the marginal or not. Equipped with the statistics of the source terms, plugging the rBP equations (A.14)-(A.17) into the order parameter definitions (D.1)-(D.3) and using self-averaging as in the example (D.5), we obtain the state evolution equations:

$$M_w^{z,t} = \mathbb{E}_{w_z^0, J_z^{w,t}} \left[ f_{\text{in}}^w(K_z^{w,t}, J_z^{w,t}) \, w_z^0 \right] \tag{D.24}$$

$$M_v^{z,t} = \mathbb{E}_{(v_X^0, v_Y^0), J_X^{v,t}, J_Y^{v,t}} \left[ f_{\text{in}}^{v^z}(K_X^{v,t}, K_Y^{v,t}, J_X^{v,t}, J_Y^{v,t}) \, v_z^0 \right] \tag{D.25}$$

$$Q_w^{z,t} = \mathbb{E}_{w_z^0, J_z^{w,t}} \left[ f_{\text{in}}^w(K_z^{w,t}, J_z^{w,t})^2 \right] \tag{D.26}$$

$$Q_v^{z,t} = \mathbb{E}_{(v_X^0, v_Y^0), J_X^{v,t}, J_Y^{v,t}} \left[ f_{\text{in}}^{v^z}(K_X^{v,t}, K_Y^{v,t}, J_X^{v,t}, J_Y^{v,t})^2 \right] \tag{D.27}$$

$$\Sigma_w^{z,t} = \mathbb{E}_{w_z^0, J_z^{w,t}} \left[ \frac{\partial f_{\text{in}}^w}{\partial J}(K_z^{w,t}, J_z^{w,t}) \right] \tag{D.28}$$

$$\Sigma_v^{z,t} = \mathbb{E}_{(v_X^0, v_Y^0), J_X^{v,t}, J_Y^{v,t}} \left[ \frac{\partial f_{\text{in}}^{v^z}}{\partial J_{1/2}}(K_X^{v,t}, K_Y^{v,t}, J_X^{v,t}, J_Y^{v,t})^2 \right] \tag{D.29}$$

with scalars $w_z^0 \sim P_{w_z^0}$ and $(v_X^0, v_Y^0) \sim P_v$ and the source terms

$$J_z^{w,t} \sim \mathcal{N}\left( \frac{\alpha_z \lambda_z \lambda_z^0}{\hat{\Delta}^z} M_v^{z,t} w_z^0, \ \frac{\alpha_z \lambda_z^2}{\tilde{\Delta}^z} Q_v^{z,t} \right) \tag{D.30}$$

$$J_z^{v,t} \sim \mathcal{N}\left( \frac{\lambda_z \lambda_z^0}{\hat{\Delta}^z} M_w^{z,t-1} v_z^0, \ \frac{\lambda_z^2}{\tilde{\Delta}^z} Q_w^{z,t-1} \right) \tag{D.31}$$

$$K_z^{w,t} = \alpha_z \frac{\lambda_z^2}{\tilde{\Delta}^z} Q_v^{z,t} - \alpha_z \lambda_z^2 \bar{R}^z (Q_v^{z,t} + \Sigma_v^{z,t}) \tag{D.32}$$

$$K_z^{v,t} = \frac{\lambda_z^2}{\tilde{\Delta}^z} Q_w^{z,t-1} - \lambda_z^2 \bar{R}^z (Q_w^{z,t-1} + \Sigma_w^{z,t-1}). \tag{D.33}$$

Note that the distributions of $J_z^{w,t}, J_z^{v,t}$ still depend on $w_z^0, v_z^0$, therefore the average is performed also over the priors in (D.26) - (D.29). In these 12 equations, all variables are scalars, giving the low-dimensional effective description of the relaxed BP as well as the AMP dynamics.

### D.1 Bayes-optimal priors and Gaussian noise channels

The general state evolution depends on the noise channels through $\hat{\Delta}^z, \tilde{\Delta}^z$, and $\bar{R}^z$. Using that for the Gaussian channel $P_{\text{out}}(z_{ij}, w_i^z, v_j^z) = \mathcal{N}(\lambda_z \frac{w_i^z v_j^z}{\sqrt{n_z}}, \sigma_{\xi^z}^2)$ we have $S_{ij}^z = \frac{z_{ij}}{\sigma_{\xi^z}^2}$ and $R_{ij}^z = \frac{z_{ij}^2}{\sigma_{\xi^z}^4} - \frac{1}{\sigma_{\xi^z}^2}$, we find by plugging into (D.11), (D.16) and (D.19) that

$$\hat{\Delta}^z = \tilde{\Delta}^z = \sigma_{\xi^z}^2 \tag{D.34}$$

$$\bar{R}^z = 0. \tag{D.35}$$

Furthermore, for Bayes-optimal priors we can use the Nishimori identity (21) if the state evolution follows the Nishimori line (Nishimori, 2001). Due to the symmetry spontaneously broken at initialization, as discussed in Section 2.4, $f_{\text{in}}^x(K_i, J_i)$ is the mean of the local posterior distribution $\mathcal{W}(x, K_i, J_i)$ with broken symmetry estimating $\pm x_i^0$, depending on the sign of the node average $\mathbb{E}_i[x_i^0 f_{\text{in}}^x(K_i, J_i)]$. Conditioned on the $\pm$ direction of broken symmetry, $f_{\text{in}}^x(K, J|\pm)$ has nonzero mean so that self-averaging (D.4) applies and node and disorder average can be exchanged. Then we have the not obvious application

$$\mathbb{E}_{x^0}[(f_{\text{in}}^x(K, J|\pm))^2] = \mathbb{E}_{x^0}[\mathbb{E}_{\mathcal{W}_{K,J,\pm}}(x)\mathbb{E}_{\mathcal{W}_{K,J,\pm}}(x)] \tag{D.36}$$

$$= \mathbb{E}_{x^0}[\mathbb{E}_{x_1, x_2 \sim \mathcal{W}_{K,J,\pm}}(x_1 x_2)] \tag{D.37}$$

$$= \mathbb{E}_{x^0}[\mathbb{E}_{x_2 \sim \mathcal{W}_{K,J,\pm}}(\pm x^0 x_2)] \tag{D.38}$$

$$= \pm \mathbb{E}_{x^0}[x^0 \ f_{\text{in}}^x(K, J|\pm)] \tag{D.39}$$

$$= |\mathbb{E}_{x^0}[x^0 \ f_{\text{in}}^x(K, J)]|, \tag{D.40}$$

where $K, J$ of course depend on $x^0$ and in the last step we could exchange the $\pm$ condition for the absolute value. Thus (D.26) and (D.27) yield

$$Q_w^{z,t} = |M_w^{z,t}| \tag{D.41}$$

$$Q_v^{z,t} = |M_v^{z,t}|. \tag{D.42}$$

The Nishimori identity can also be applied to $Q_{w/v}^{z,t} + \Sigma_{w/v}^{z,t}$, since

$$\mathbb{E}_{x^0}[(f_{\text{in}}^x(K, J))^2 + \partial_J f_{\text{in}}^x(K, J)] = \mathbb{E}_{x^0}[\mathbb{E}_{\mathcal{W}_{K,J}}(xx)] = \mathbb{E}_{x^0}[x^0 x^0], \tag{D.43}$$

so for priors without mean

$$Q_{w/v}^{z,t} + \Sigma_{w/v}^{z,t} = \sigma_{w^z/v^z}^2 \tag{D.44}$$

The last relation is not needed for the Gaussian channel case, as the terms involving $\Sigma_{w/v}^{z,t}$ vanish anyways due to $\bar{R}^z = 0$, (D.35). In total, using (D.34),(D.35),D.41, the state evolution simplifies to the form given in Equations (22) and (23).

### D.2 Fully Gaussian, and Rademacher-Bernoulli models

With Bayes-optimal, Gaussian priors and Gaussian noise channels, the expectations over both the prior and the source term give a simple closed form. We use the short hands $\tilde{M}_{w/v}^{z,t} = \frac{\lambda_z^2}{\sigma_{\xi^z}^2} M_{w/v}^{z,t}$ as introduced also below (22). The denoising function (B.11) being a linear function in $J$, the average over $J_z^{w,t}$ in (23) is given by the respective mean $\mathbb{E}_{J_z^{w,t}}[J_z^{w,t}]$, leading to

$$\mathbb{E}_{J_z^{w,t}}\left[f_{\text{in}}^w\left(\alpha_z|\tilde{M}_v^{z,t}|, J_z^{w,t}\right)\right] = \frac{\alpha_z \tilde{M}_v^{z,t} w_z^0}{\alpha_z |\tilde{M}_v^{z,t}| + \sigma_{w^z}^{-2}} \tag{D.45}$$

and $f_{\text{in}}^{v^z}$ is again linear in $J_X, J_Y$, so the average over $J_z^{v,t}$ yields

$$\mathbb{E}_{J_X^{v,t}, J_Y^{v,t}}\left[f_{\text{in}}^{v^z}\left(|\tilde{M}_w^{X,t-1}|, |\tilde{M}_w^{Y,t-1}|, J_X^{v,t}, J_Y^{v,t}\right)\right] = v_z^0 \tilde{M}_w^{z,t-1} \tilde{\Sigma}_{zz}\left(|\tilde{M}_w^{X,t-1}|, |\tilde{M}_w^{Y,t-1}|\right)$$
$$+ v_{\bar{z}}^0 \tilde{M}_w^{\bar{z},t-1} \tilde{\Sigma}_{z\bar{z}}\left(|\tilde{M}_w^{X,t-1}|, |\tilde{M}_w^{Y,t-1}|\right). \tag{D.46}$$

Then also the averages over the prior distributions in (23),(22) simplify to $\mathbb{E}_{w_z^0}[(w_z^0)^2]$, $\mathbb{E}_{v_z^0}[(v_z^0)^2]$ and $\mathbb{E}_{v_z^0}[v_z^0 v_{\bar{z}}^0]$, so the SE equations are

$$M_v^{z,t} = \sigma_{v^z}^2 \tilde{M}_w^{z,t-1} \tilde{\Sigma}_{zz} \left( |\tilde{M}_w^{X,t-1}|, |\tilde{M}_w^{Y,t-1}| \right)$$
$$+ c_v \tilde{M}_w^{\bar{z},t-1} \tilde{\Sigma}_{z\bar{z}} \left( |\tilde{M}_w^{X,t-1}|, |\tilde{M}_w^{Y,t-1}| \right) . \tag{D.47}$$

$$M_w^{z,t} = \frac{\alpha_z \tilde{M}_v^{z,t} \sigma_{w^z}^2}{\alpha_z |\tilde{M}_v^{z,t}| + \sigma_{w^z}^{-2}} \tag{D.48}$$

When changing to a Rademacher-Bernoulli (sparse) prior on $w^z$ while $v^X, v^Y$ remains jointly Gaussian, Equation (D.47) remains the same. The expectation over the sparse prior in the $M_w^{z,t}$ update (23) simply gives a sum of three terms which is omitted here for brevity. Then only the Gaussian integral over the source term $J_z^{w,t}$ must be computed numerically.

## D.3 Bethe free energy in general case

Based on the form of the SE equations in Appendix D and in analogy to the replica calculation of Lesieur et al. (2017) in their Appendix C, we read off the Bethe free energy, which corresponds to the free energy obtained by a replica-symmetric Ansatz, and which we state here without lengthy derivation

$$\Phi^{RS}(\{M, Q, \Sigma\}) = \sum_{\{z\}} \left( \frac{\lambda_z \lambda_z^0}{\hat{\Delta}^z} M_{w^z} M_{v^z} - \frac{\lambda_z^2}{2\tilde{\Delta}^z} Q_{w^z} Q_{v^z} \right)$$
$$+ \sum_{\{z\}} \left( \lambda_z^2 \bar{R}^z (Q_w^z + \Sigma_w^z)(Q_v^z + \Sigma_v^z) \right)$$
$$- \sum_{\{z\}} \frac{1}{\alpha^z} \mathbb{E}_{w_z, J_z^w} \left[ \log \mathcal{Z}_w(K_z^w, J_z^w) \right] \tag{D.49}$$
$$- \mathbb{E}_{(v_X^0, v_Y^0), J_X^v, J_Y^v} \left[ \log \mathcal{Z}_v(K_X^v, K_Y^v, J_X^v, J_Y^v) \right] .$$

Here $\mathcal{Z}(K, J)$ are the normalizations of the tilted priors $\mathcal{W}(K, J)$ defined in (10). We can interpret the last two lines of (D.49) as the energetic terms and the first two lines as the additional entropic contributions arising from the introduction of the order parameters after integrating out the Fourier variables. The relation to state evolution is that the stationarity condition

$$\vec{\nabla}_{\{M, Q, \Sigma\}} \Phi^{RS} \overset{!}{=} 0 \tag{D.50}$$

gives back exactly the SE equations (D.24)-(D.33).

## D.4 Bethe free energy for Rademacher-Bernoulli prior and Gaussian channels

The Bethe free energy (D.49) simplifies to

$$\Phi^{RS}(\{M\}) = \sum_{\{z\}} \frac{1}{2} M_{w^z} \tilde{M}_{v^z} - \sum_{\{z\}} \frac{1}{\alpha^z} \overline{\log \mathcal{Z}_{w^z}} - \overline{\log \mathcal{Z}_v} \tag{D.51}$$

where the free energy of the Gaussian part can be computed analytically, with $K_z = |\tilde{M}_{w^z}|$ as well as $J_z \sim \mathcal{N}\left(\tilde{M}_{w^z} v_0^z, |\tilde{M}_{w^z}|\right)$, and therefore

$$\overline{\log \mathcal{Z}_v} = \mathbb{E}_{(v_0^X, v_0^Y), J_X, J_Y} \left[ \frac{1}{2} \log \det \tilde{\Sigma}_K + \frac{1}{2} J^T \tilde{\Sigma}_K J \right] \tag{D.52}$$

$$= \frac{1}{2} \log \det \tilde{\Sigma}_K + \frac{1}{2} \sum_{\{z\}} (\tilde{M}_{w^z}^2 \sigma_{v^z}^2 + |\tilde{M}_{w^z}|) \tilde{\Sigma}_K^{zz} + \frac{1}{2} \tilde{M}_{w^X} \tilde{M}_{w^Y} c_v (\tilde{\Sigma}_K^{XY} + \tilde{\Sigma}_K^{YX}). \tag{D.53}$$

The free energy of the Rademacher-Bernoulli part yields in turn with $K_z = \alpha^z |\tilde{M}_{v^z}|$ as well as $J_z \sim \mathcal{N}\left(\alpha^z \tilde{M}_{v^z} w_0^z, \alpha^z |\tilde{M}_{v^z}|\right)$, the expression

$$\overline{\log \mathcal{Z}_{w^z}} = \mathbb{E}_{w^z, J_z}\left[\log(\rho_z \cosh(J_z) e^{-\frac{1}{2}K_z} + 1 - \rho_z)\right], \tag{D.54}$$

where the sum over the three states of the Rademacher-Bernoulli prior can be written out straightforwardly, which we omit here for brevity.

## E Supplementary figures

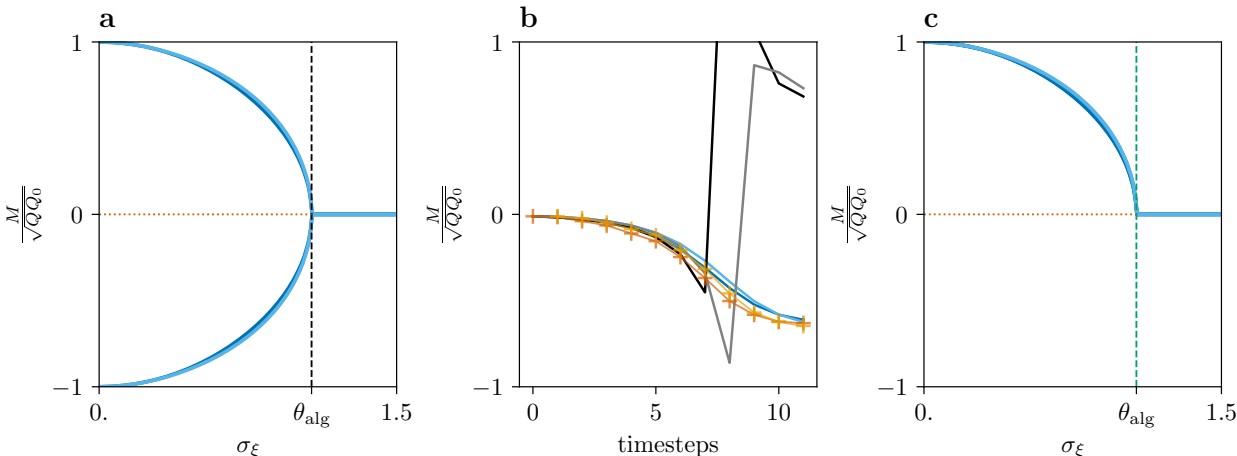

Figure S1: Introducing $Q = |M|$ fixes asymmetry of state evolution. **a** The fully symmetric branches of the phase transition for the Gaussian model without squaring the cosine similarities. Here $Q_z = |M_z|$ according to Equations (22) and (23), and parameters as in Figure 2. At $\theta_{\text{alg}}$ the uninformative fixed point looses stability (orange dotted) and two stable informative branches exist, representing the $\mathbb{Z}_2$ symmetry. **b** Time resolved trajectory of the cosine similarities $S_{C,w^x}$ and $S_{C,v^x}$, starting from a random vector with negative overlaps. The trajectory of AMP (orange '+') is consistent with the prediction of symmetric SE (blue lines), while the prediction of asymmetric SE based on $Q_z = M_z$ (grey lines) is not physical. Parameters as in **a** with $\sigma_{\xi^z} = 0.8$, and the AMP trajectory is one run at $d = 10000$. **c** The branches of SE if $Q_z = M_z$. The branch of fixed points with negative overlaps does not exist, only the branch with positive overlaps is stable.

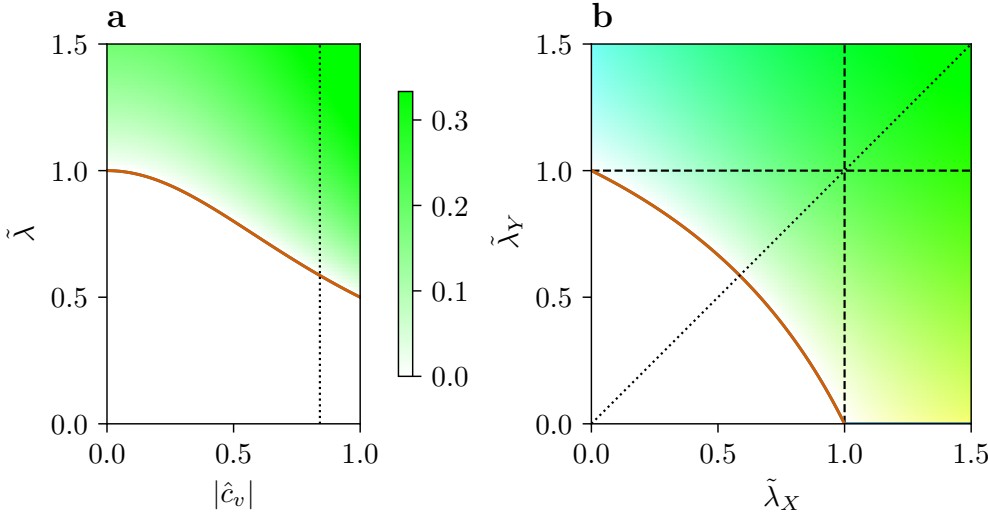

Figure S2: Phase diagram of weak recovery threshold as in Figure 5, but also showing Bayes-optimal performance specific to the model with Gaussian priors and Gaussian noise channel. The product of cosine similarities, $CS_{w^z}CS_{v^z} = \frac{M_{w^z}M_{v^z}}{\sigma_{w^z}\sigma_{v^z}\sqrt{Q_{w^z}Q_{v^z}}x}$ as obtained from SE (23,22) is shown. **a** Both modalities are symmetric, $CS_{w^z}CS_{v^z}$ is indicated by the green color scale. **b** The performance achievable in the two modalities differs. $CS_{w^X}CS_{v^X}$ is shown in blue and $CS_{w^Y}CS_{v^Y}$ in yellow, mixing to the green color scale on the diagonal which corresponds to the color bar given in panel **a**. Again the dotted lines indicate the intersection of the two planes.

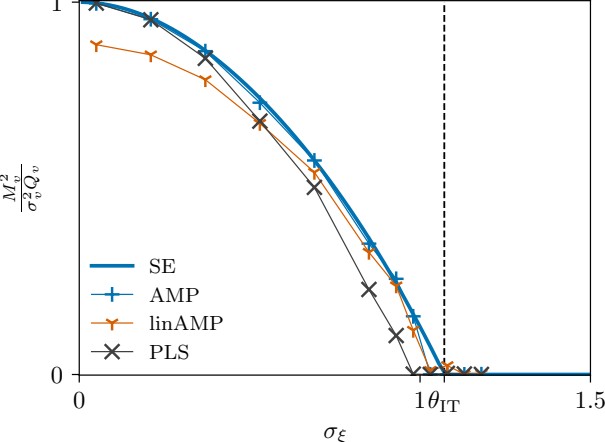

Figure S3: As Figure 2**a**, but showing the performance of estimating $v^z$ from the same simulations. The $v^z$ estimate of linearized AMP in the regime of small noise is not perfect since the operator performs a weighted average of the $v^X$ and $v^Y$ estimates, as discussed at the end of Section 2.2.

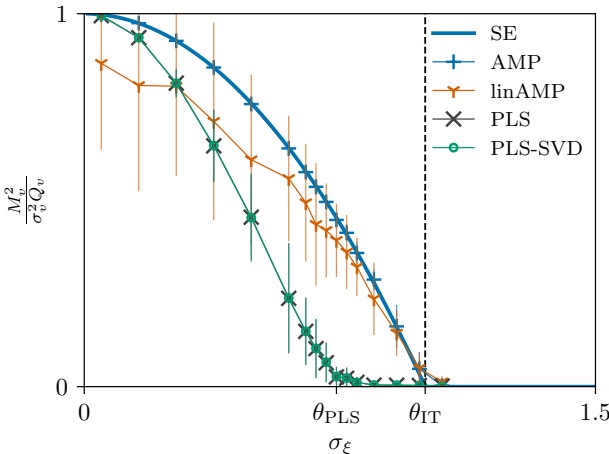

Figure S4: As Figure 4, but showing the performance of estimating $v^z$ from the same simulations. Notably, for $v^z$ there is no difference between PLS-Canonical and PLS-SVD, since the additional regression step distinguishing the two is to estimate $w^z$. The $v^z$ estimate of linearized AMP in the regime of small noise is again not optimal and shows a larger variance than the estimate of $w^z$ in Figure 4.

