# OpenReview forum: "Optimal thresholds and algorithms for a model of multi-modal learning in high dimensions"
_TMLR — Rejected by TMLR_

### Review · Reviewer_2L6r · 2024-10-21

**Summary Of Contributions:**

This paper aims to evaluate the performance of the Approximate Message Passing (AMP) algorithm for a rank-1 model with Gaussian additive noise. The contribution seems to characterize the behavior of AMP under various conditions, providing some evaluation of its performance.

**Audience:**

No

**Claims And Evidence:**

No

**Requested Changes:**

1. Could you please clarify what the real *technical contribution* of this paper is? Is it simply an analysis of an existing algorithm on some specific models?

2. Could you please rewrite the paper in a Definition-Theorem-Proof format, so that a general machine learning researcher can better understand it?

**Strengths And Weaknesses:**

**Disclaimer:** This paper is outside my area of expertise, so I can only offer some general comments.

**Weaknesses:**

1. The paper is very difficult to read. It is nearly impossible to understand what the paper is attempting to achieve and what its true technical contributions are.

2. The writing resembles more of a technical report, focusing heavily on derivations without providing motivation, intuition, or discussion about why these derivations matter.

Unless I have missed some critical aspects, I believe the lack of clarity in the presentation alone warrants a rejection at TMLR.

---

### Review · Reviewer_Scht · 2024-10-21

**Summary Of Contributions:**

The authors explore the matrix estimation problem within a multi-modal framework, specifically focusing on a rank-1 matrix estimation task with correlated latent spaces in a dual-view setting. Their primary goal is to analytically quantify the performance improvements achieved through multi-modal inference compared to analyzing each modality in isolation. To this end, they offer the following key contributions:

1. They provide information-theoretic performance limits for their simplified setting.
2. Given the prior information about the covariance between the latent vectors, they provide a quantitative analysis of the signal-to-noise gain from the optimal combination of two views.
3. Through extensive numerical experiments, they compare their proposed method, based on Approximate Message Passing (AMP), against standard approaches like Partial Least Squares (PLS) and Canonical Correlation Analysis (CCA).

**Audience:**

Yes

**Claims And Evidence:**

Yes

**Requested Changes:**

- The work by Lesieur et al. (2017) appears to have significantly influenced the proposed approach, particularly in the adoption of ideas and analysis related to Approximate Message Passing (AMP). It would be helpful if the authors could compare and contrast the key technical contributions of their work with those of Lesieur et al. (2017). Additionally, it would be insightful to understand the challenges the authors encountered when extending the methods from Lesieur et al. (2017) to the dual-view setting.
- While the simplified rank-1 model captures several interesting phenomena, it would be helpful if the authors could discuss the key challenges involved in extending their method to a more general rank-$r$ model.

**Strengths And Weaknesses:**

Using a simplified rank-1 model, the authors draw important insights into multi-modal inference and highlight key differences from single-view methods. Notably, their analysis of algorithmic and information-theoretic weak recovery thresholds is an interesting contribution.

The paper is densely written and relies heavily on notation, which can make it challenging to follow at times. Some notations are cumbersome or undefined, adding to the difficulty. The authors should ensure that abbreviations like the BBP transition - perhaps familiar within the niche community but not widely known - are clearly defined. Similarly, notations such as $\overset{!}{=}$ in Eqs. (27, 28, 29) appear without prior explanation, which may confuse readers. Overall, readers unfamiliar with the field may find the paper difficult to follow.

---

### Review · Reviewer_MAQE · 2024-10-23

**Summary Of Contributions:**

The authors analyse inference in a linear multi-view, or multi-modal, matrix factorisation problem, where there are two views (or modalities) and the matrices are rank 1 (up to noise). The authors provide asymptotic thresholds (phase transitions) for Bayes-optimal inference and algorithmic thresholds for message passing (weak recovery thresholds). They validate these thresholds numerically, and compare to standard methods (PLS, CCA) which are shown to be suboptimal, in contrast to the single-view case.

**Audience:**

Yes

**Claims And Evidence:**

No

**Requested Changes:**

Each requested change is set out alongside the weaknesses above.

**Strengths And Weaknesses:**

I did not follow carefully all of the theoretical and methodological details, and so am not sure of the correctness of the main claims.

Strengths:
* (major) The evaluation is clear. It appears to me that the numerical evaluation is well-chosen.

Weaknesses:
* (major) The motivation for some of the choices is not clear. In particular the authors explicitly claim that a rank-1 model is enough to capture the phenomenology of the problem, and also only consider two modes. This could be improved by, for example, providing a more detailed explanation of the authors' intuition, providing some empirical studies to support the claim, or by being more agnostic as to whether this case captures the behaviour of a wider class.
* (major) some elements of the paper could be more clearly written, particularly if it is to be readable by a broader audience. For example, it would be helpful to have a running example application in the introductory section. Also, it would be helpful to state the thresholds found as a formal result (theorem, or similar), clearly setting out the assumptions and the method off derivation.

Overall, I think the paper contains some quite interesting results, but I would not be able to comment on their correctness, and some improvement could be made to the writing to make the paper shine more.

---

### Review · Reviewer_ezsD · 2024-10-23

**Summary Of Contributions:**

*Disclaimer*: The topic is outside of my expertise. I have limited ability to judge the correctness and contribution

This paper aims to quantify the theoretical gain of multimodal inference. In particular, the authors studied a simplified rank-1 model with Gaussian additive noise and two views / modalities. The authors derived an approximate message passing (AMP) algorithm to characterize the performance in the high dimensional limits, successfully quantifying the SNR gain from optimally combining two views with respect to different levels of correlations between the latents of the two modalities. Empirically, the authors also showed that the derived algorithm overcomes limitations from alternative methods like PLS and CCA by combining the information from individual and correlated views.

**Audience:**

Yes

**Claims And Evidence:**

Yes

**Requested Changes:**

See the weaknesses above

**Strengths And Weaknesses:**

Strengths
* Clear derivation of the methods and clear explanations of why certain design choices are made. For example, choosing to update messages into blocks to avoid problems arriving from Z_2 symmetry.
* Clear explanation in the numerical result section of where and why PLS, CCA underperforms
* Discussion of special case (no correlation / perfect correlation)

Weaknesses
* The paper is very dense in technical terms, with many of them unexplained (eg Nishimori identity,  Onsager reaction term). This makes it hard of general ML readers to follow.
* Rank-1 and two-modality are very simplified from a practical point of view. It is unclear to me how trivial it is to extend the conclusions beyond the setup.

---

### Review · Reviewer_YAHU · 2024-10-23

**Summary Of Contributions:**

This paper studies multi-modal inference under rank-1 simplified model using AMP algorithm. The problem model is rank-1 model with Gaussian noise, where independent $w^X$ and $w^Y$ are multiplied to the correlated vectors $v^X$ and $v^Y$. Specifically, $X=w^X (v^X)^\top + N^X$ and $Y=w^Y (v^Y)^\top + N^Y$ with some noise $N^X$ and $N^Y$. To infer the posterior probability $P(w,v|X,Y)$, the authors derive AMP for the rank-1 model, and analyze the high-dimensional limit using state evolution. Linearized AMP is also provided and is compared with PLS and CCA methods. Through numerical simulation, the authors provide some findings about the gain from multi modal learning.

**Audience:**

Yes

**Claims And Evidence:**

No

**Requested Changes:**

1. The paper should state its objective, the goal of the rank-1 model, motivation, and intuition. For example, I didn't know the goal is to find posterior probability during reading Section1.1. In particular, after (4) the authors stated that we aim to analyze Bayes-optimal estimation, but what to estimate?
2. Some terminologies (BBP transition, dynamics, Bethe free energy, Nishimori identity, Onsager reaction, $\overset{!}{=}$, etc) are not defined, so please define or explain terms if it is not introduced.
3. Could the authors summarize the main contribution regarding multimodal learning? It seems that the authors compares algorithms on the simplified model. Is the main contribution is deriving AMP on the rank-1 model?

**Strengths And Weaknesses:**

Strengths.

Analysis of theoretical gain of using multimodal data is important in the multimodal learning. Even through the model considered in the paper is too simple, I think the model is enough to capture statistical gain of using multi modalities.
The authors also derive AMP algorithm for this model, and provide findings by numerically comparing it with linearized AMP, CCA, and PLS.

Weaknesses.

In my opinion, the paper is quite difficult to read as lots of contents are stated without intuition and motivation. It would be better to make the paper readable to broader ML researchers especially not familiar with AMP. Moreover, some terminologies are used without defining such as BBP transition, dynamics of rBP, etc. Moreover, I am not sure what are the core findings regarding the use of multimodal dataset. Section 3 provides some discussion on multimodality based on numerical simulation with several algorithms, but does comparing algorithms provide meaningful analysis on multimodal view learning? It would be easier to read if the paper concisely presents core findings (about multimodal gain) clearly.

---

### Author Response · Authors · 2024-10-23
**Revision and reply, many thanks for the reviews**

We would like to thank reviewers YAHU, ezsD, MAQE, Scht, and 2L6r very much for their reviews, and their time invested even though for some of them our manuscript was outside of their area of expertise. We synthesize here our reply and the changes we have made to the manuscript.

Overall, we thank the reviewers for their in sum positive assessment. While some concerns where raised about the heavy notation/terminology and specialized nature of our manuscript, we emphasize that none of the reviews has found a technical mistake or any problem with the evidence we provide to support our claims. Also, except for reviewer 2L6r, 4 of the 5 reviewers agree that there is an audience of readers for the manuscript.
Therefore, we believe that the acceptance criteria of TMLR (technical correctness, accurate and clear evidence for claims, and an audience for the paper) are clearly satisfied.

---

> ### Author Response · Authors · 2024-10-23
> **continuation**
>
> We now respond to the issues pointed out by the reviewers and describe the changes made to the manuscript.
>
> 1) Heavy notation and missing definitions
>
> We understand the point that the manuscript notation is heavy, and we have done our best to make the notation as simple and logical as possible in order to make the underlying structure of the equations apparent and avoid additional clutter.
> Thank you for pointing to the missing definitions. In the updated version of the manuscript we have:
> - corrected the missing definition of the Baik-BenArous-Péché (BBP) transition abbreviation,
> - clarified the $\overset{!}{=}$ notation which is common, but not ubiquitous, in physics to signify that the equality must hold at the point of interest (here the phase transition).
> - Relaxed belief propagation (rBP) is defined above eq.10.
> If there are additional missing or unclear definitions which the referees have noted, we are happy to address these.
>
>
> 2) Technical contributions
>
> Firstly we would like to clarify that our main contributions, as summarized in section 1.3,  are not on the technical/methodological side, but that we have applied existing methods to a novel problem and model to generate insight.
>
> AMP and state evolution are well-known methods to perform and analyze approximate inference in high-dimensions, we cite this literature in the introduction and the related work section. We derive and analyze the corresponding equations for the multi-view inference model which we introduce, enabling a characterization of the phase diagram and the Bayes-optimal performance limits on this task. We perform numerical experiments confirming our theoretical results. Comparing to PLS and CCA, standard methods to discover correlated latent structure in multi-view data, we find that our high-dimensional task CCA performs much worse than what is possible by Bayes-optimal estimation of the signal, and also than what is achieved by PLS.
>
> Thanks for raising the point that the degree of methodological contribution was not clear. The main technical complication in extending the work of Lesieur et al. (2017) was to incorporate the correlation between the two rank-1 signals in the Belief Propagation messages. This is discussed below eq.9 and visually represented in Fig 1. We have also added the following passage at the beginning of Sect. 2.1:
>
> > "The main technical contributions here are the formulation of a parsimonious multi-view model in \cref{sec:model} and the treatment of correlated latent variables by two-dimensional marginals in the BP messages. The remaining derivation of AMP and the state evolution then goes through as a straight-forward generalization of the calculations for single-view matrix factorization presented by \cite{Lesieur2017}, whose notation we adapt slightly for more consistency with standard symbols in statistical physics."
>
>
> 3) Statistical physics terminology and presentation without theorem-proof structure
>
> While we appreciate the suggestion to reduce the use of statistical physics terminology and present results in a theorem-proof format suitable for rigorous math, this paper is a specialized contribution whose main audience will be familiar with the key concepts in statistical physics of inference and learning, such as belief propagation or the Bethe approximation. Given TMLR’s broad scope, including specialized works, we believe the current style is appropriate for conveying the results in a manner most accessible to researchers in this field. References to the fundamental literature (Mézard \& Montanari, 2009; Zdeborová \& Krzakala, 2016) are pointed out in the introduction, concepts such as the Nishimori identity, eq.21, are briefly explained and references to the original literature are given. We are open to clarifying any specific points that may improve accessibility without altering the overall structure.
>
>
> 4) Discussion of a model extension to rank-r
>
> Below eq.5 we have added the following discussion:
>
> > "An extension to finite rank $r$ would, in analogy to single-view matrix factorization (Rangan \& Fletcher,2012; Lesieur et al., 2017), yield an additional index in the equations while the location of the phase transition for the strongest signal direction will not change. Qualitatively different behavior could appear in other scaling limits, e.g. if the signal rank is not finite but proportional to  $n_z$ and $d$."

---

### Decision · Action_Editor_jTQH · 2024-11-25

**Recommendation:** Reject

**Comment:**

Whilst I do acknowledge it is possible that the paper may have merit and may even be very good work in the eyes of a physicist (and the reviewers and I may not have been able to grasp the points clearly enough), I cannot override the opinion of the reviewers without clear cause. **All five reviewers complain of similar issues and most argue for rejection**, demonstrating that the paper is not suited to a machine learning audience in its current form. **Three reviewers recommended rejection (one of them strongly), with only one reviewer (very unenthusiastically) recommending acceptance**.

I would like to further assure the authors that I was aware of the possibility of difficulties understanding the related literature at the reviewer assignment stage and **did try hard to find appropriate reviewers**: whilst I wasn’t able to find anyone with a strong statistical physics background from the available TMLR reviewer pool, the reviewers are **competent members of the** theoretical machine learning **community** with a high affinity score for this paper and strong backgrounds in learning theory, Bayesian methods and information theory evidenced by numerous high profile publications. Therefore, I think **their complaints should be taken seriously**.

If the authors decide to re-submit a major revision to TMLR, I suggest “translating” the paper in machine learning language: the following main “building blocks” should be clearly identified in self-contained theorem/assumption/remark environment:

1. What is observed
2. What the algorithm is
3. What the performance measure is
4. The main theorem in terms of an upper bound on the performance measure
5. How the gain in SNR varies with the correlation matrix of $v^X$ and $v^Y$
6. All assumptions


In addition, the difference, both algorithmic and theoretical, with [1] should be explained clearly. Despite the authors’ assurance that the only issue with the paper is a difference in culture between the physics and machine learning literature, I cannot help but imagine it would be possible to explain the theoretical contributions more intelligibly. The problem studied isn’t far from problems studied by machine learning works. For instance, “matrix denoising” (see the beginning of section 2 and section 4 in [2]) is essentially the same problem as the one presented here, except without the multi modal aspect. The theoretical results and the assumptions required in [2] are very easy to understand. It does seem that it should be possible to make the main points of this paper (if not all the details of the algorithm) clearer to the target audience of TMLR.







Minor typos:

Pages 4, 8 and others, “iid.” should be “i.i.d.”.

***References***


[1] T Lesieur, F Krzakala, L Zdeborová, “Constrained low-rank matrix estimation: Phase transitions, approximate message passing and applications”

[2]Yuxin Chen, Yuejie Chi, Jianqing Fan, Cong Ma, “Spectral Methods for Data Science: A Statistical Perspective”

**Audience:**

I believe (generally in disagreement with all reviewers) that the **problem studied**, i.e. recovering the factors of two correlated low-rank matrices based on a noisy observations **is of interest to the community**.  However, the description of the results are not clear enough to appeal to a machine learning audience.

**Claims And Evidence:**

This paper studies the problem of Bayes optimal multi-view rank 1 matrix denoising: we fully observe a noisy version of two matrices $X=w^X(v^X)^\top$ and $Y=w^Y(v^Y)^\top$, where $v^X$ and $v^Y$ are assumed to be correlated, whilst $w^X$ and $w^Y$ are assumed to be independent. The ground truth matrices are recovered with an approximate message passing algorithm, and the gains in signal to noise ratio are characterised as opposed to the situation where only a single matrix is observed (which was studied extensively in [1] for an arbitrary rank $r$).



Whilst it appears the contributions could be significant to a statistical physics audience, **all five reviewers** independently **complained** of the **lack of readability** of the work to a machine learning audience.
There are plenty of notations which are not introduced, and the paper generally assumes the reader is fully familiar with statistical physics literature. After attempting to read the paper and rebuttal myself, I have to agree with the reviewers on that point. For instance, notations such as the $g_z$ in equation (3) or terms such as “Onsager reaction term”  or “Nishimori identity” introduced unceremonially and without proper background. Even the observation model takes a bit of time to  guess from the writing of the paper: I understand now that $X,Y$ are both fully observed, but the analogy with the matrix factorization literature (where only a subset of the entries would be observed) somewhat obscures this fact, and the fact is not really explicitly stated in the relevant section 1.1, which simply states “we consider the following rank 1 model: $X=…$, $Y=…$. Something I find even harder to handle is that the writing style is like a narrative and doesn’t delineate the assumptions very cleanly. For instance, the underlying assumption behind equation (5) is sprinkled in the middle of the narrative. It is not stated that the vectors $w^X$ and $w^Y$ are Gaussian, only that they are independent. It is not clearly explained why the performance measure can be expressed in terms of the message passing functions $M$ as is done here. Similarly, the final results are also not clearly delineated. What is the main theorem? It appears there is none as the authors mention in the rebuttal that “the contributions….are not on the technical/methodological side”. Yet the authors do claim that they have shown improved SNR compared to the single view situation, but if that is a rigorous theoretical contribution, it is not explained anywhere. If it is purely an experimental observation, it should be made clearer as well. As a reviewer points out in his final recommendation, such **problems persist in the revised version** more recently uploaded by the authors.


The authors' statement, added after the rebuttal stage that "An extension to finite rank r would, ....would yield an additional index in the equations while the location of the phase transition for the strongest signal direction will not change." seems quite bold and unsupported by concrete claims.

**Resubmission Of Major Revision:**

The authors may consider submitting a major revision at a later time.